

# Constructing the infrared conformal generators on the fuzzy sphere

**Giulia Fardelli\*, Andrew Liam Fitzpatrick† and Emanuel Katz‡**

Department of Physics, Boston University, Boston, MA 02215, USA

\* fardelli@bu.edu , † fitzpatr@bu.edu , ‡ amikatz@bu.edu

## Abstract

We investigate the conformal algebra on the fuzzy sphere, and in particular the generators of translations and special conformal transformations which are emergent symmetries in the infinite IR but are broken along the RG flow. We show how to extract these generators using the energy momentum tensor, which is complicated by the fact that one does not have a priori access to the energy momentum tensor of the CFT limit but rather must construct it numerically. We discuss and quantitatively analyze the main sources of corrections to the conformal generators due to the breaking of scale-invariance at finite energy, and develop efficient methods for removing these corrections. The resulting generators have matrix elements that match CFT predictions with accuracy varying from sub-percent level for the lowest-lying states up to several percent accuracy for states with dimension $\sim 5$ with $N = 16$ fermions. We show that the generators can be used to accurately identify primary operators vs descendant operators in energy ranges where the spectrum is too dense to do the identification solely based on the approximate integer spacing within conformal multiplets.



# 1   Introduction and summary

High-energy states of Quantum Field Theory (QFT) and their various properties remain a largely unexplored terrain. Such states are very interesting, however, as they are believed to offer important insights into chaos, thermalization, and the emergence of nontrivial phases of matter including hydrodynamics and superfluids among others [1–4]. In the case of conformal field theories (CFTs), one at least has the advantage that high-energy data is intrinsically discrete and thus, perhaps, the connection between chaos and CFTs can more easily be studied. In addition, high energy CFT data is useful in order to study more general QFTs, which can be thought of as relevant deformations of CFTs. Specifically, in the Hamiltonian Truncation framework for describing such QFTs, the more high-energy CFT data is available, the more accurately one can capture generic QFT observables [5]. Yet despite the need for such high-energy data, until recently, it has been very challenging to access high-energy CFT states using existing methods. The fuzzy sphere approach to CFT data offers a new tool that can obtain such states [6–15]. Indeed, these high-energy states can be computed numerically by diagonalizing the Hamiltonian of a dynamical system of a large number of interacting fermions in the lowest Landau level (LLL), living on a sphere in a monopole background. As the number of fermions is enlarged, with their interactions tuned appropriately to quantum criticality, one realizes an increasing number of approximate CFT states near the ground-state of the system.

A natural question which then arises is how to test that these numerically computed states approximate CFT physics rather than simply a set of interacting non-relativistic fermions. For example - how does one determine the effective UV-cutoff for the emergent IR CFT states? One way to do this is to examine the conformal structure by constructing the special conformal generators using the operators of the microscopic fermionic theory and then employ these to directly test conformality.[1] In this paper, we will take a step in this direction, by focusing on the fuzzy sphere realization of the 3$d$ Ising CFT. Our goal will be to build approximate conformal

---

[1]A similar approach has been taken in 2$d$ lattice models with a CFT fixed point [16,17], where one constructs all the generators of the conformal algebra in terms of the underlying lattice operators. See also [18] for a different construction of the 2$d$ conformal algebra.

generators and check aspects of conformality, including the conformal algebra, numerically, for the IR CFT states. We will also use the algebra as a way of extracting a rough CFT UV-cutoff from the numerical data. The hope is then that quantifying the range of emergent conformality will aid in identifying reliable high-energy CFT states for future work.

The fuzzy sphere framework has two important advantages which allow for a systematic improvement of CFT measurements: The first, is that it preserves rotational invariance perfectly, allowing one to more easily classify states and to relate microscopic fermionic operators to emergent IR CFT operators. The second, is that the interactions on the sphere are local and there is a large energy gap for single particle excitations above the LLL. Consequently, all corrections to CFT observables are also local. In other words, at criticality all CFT violations come from local irrelevant operators generated along the RG-flow to the Ising CFT. We can then significantly improve the results through the use of effective theory and conformal perturbation theory [19].

In particular, it is important to emphasize that our goal will not be to use the fuzzy sphere to independently verify the precise results of the conformal bootstrap. Rather, as we are ultimately interested in higher-energy CFT states, we are going to use the most accurate low-energy data of the conformal bootstrap in order to tune to the critical point. As we describe in detail in section 4.2, we will choose microscopic couplings in order to set both the coefficients of $\epsilon$ and $\epsilon'$ approximately to zero. However, we will find that this tuning is insufficient to obtain accurate special conformal generators. The generators are naturally constructed from the fuzzy sphere energy density local operator, but this operator near the IR CFT fixed point contains corrections from both irrelevant primary operators as well as irrelevant descendant operators. The irrelevant descendant operators do not contribute to the spectrum, but do modify the conformal generators. Therefore, we will require additional tuning in order to obtain improved generators. Our procedure for tuning is described in section 4.

Armed with these improved special conformal generators (whose matrix elements agree with CFT expectations at the few percent level for low-energy states), we will report on various detailed checks of the conformal structure. These include numerical tests of the conformal algebra, the existence of primary states (as states annihilated by the special conformal generator), as well as the spectrum of the conformal Casimir. The results are presented in sections 5.1, 5.3, and 5.4. Very roughly, we find that for $N = 16$, the conformal structure appears to break down at energies of $\sim 6$. Beyond that scale, for example, it is difficult to reliably identify primaries. The results presented here can be used to improve the fuzzy sphere program in various directions: perhaps of utmost importance is the need to push numerically to higher values of $N$ so that the cutoff is increased. This would be highly desirable for both the study of chaos as well as for Hamiltonian truncation applications, and studies of large charge EFTs of CFT states. How high exactly one must push in dimensions for these applications is difficult to predict in advance, but there are at least some examples where even $\Delta \sim 6$ is sufficient, such as the large charge EFT of the $3d$ O(2) model which appears to apply already to some states with conformal dimensions slightly larger than one [20, 21]. Relatedly, it would be useful to improve the accuracy of CFT measurements. This can be done by considering a larger space of couplings in the UV Hamiltonian. Using the larger parameter space should then allow for better tuning to criticality, for example by considering more of the $\lambda_n$ terms (defined in section 2) to set several irrelevant operator coefficients to zero. The generators can similarly be improved systematically. The result should be sensible conformal structure stretching to higher energies, indicating many more healthy states. Finally, for the above mentioned applications, it would also be advantageous to use the same tuning approach to improve any local operators, by constructing combinations of UV operators, which come closer to representing IR CFT operators. Such combinations can then be used to extract more accurate CFT OPE coefficients for excited states, starting from the results in [7].

*Comment on notation: capital letters (A, B) denote indices for the embedding space description of $S^2$, lower-case letters (a, b) denote intrinsic $S^2$ indices ($\theta, \phi$), and Greek letters ($\mu, \nu$) denote $\mathbb{R} \times S^2$ indices ($t, \theta, \phi$).*

## 2 Lightning fuzzy sphere review

The system of the lowest Landau level (LLL) at half-filling on a fuzzy sphere has been covered in many places (see e.g. [22]), and the following review will be extremely brief and is mainly to establish conventions. We restrict to the case where the action is a sum of terms that are quadratic or quartic in nonrelativistic fermion fields $\psi$:

$$
\begin{aligned}
S &= S_2 + S_4, \\
S_2 &= \int dt \, d^2x \sqrt{g} \left[ \psi^\dagger (iD_t - \frac{D^2}{2M} + h\sigma^x)\psi \right], \\
S_4 &= -\int dt \, d^2x \sqrt{g} \sum_n \left[ \lambda_n (\psi^\dagger \psi) \vec{\nabla}^{2n} (\psi^\dagger \psi) - \lambda_{n,z} (\psi^\dagger \sigma^z \psi) \vec{\nabla}^{2n} (\psi^\dagger \sigma^z \psi) \right].
\end{aligned}
\tag{1}
$$

The covariant derivative is $D_a = \nabla_a + iA_a$, where $A_a$ is a background gauge field.

We then take space to be a sphere $S^2$ with radius $R$, $ds^2 = g_{ab}dx^a dx^b = R^2(d\theta^2 + \sin^2\theta \, d\phi^2)$, and the background gauge field to be that of a magnetic monopole,

$$
A = s\cos\theta \, d\phi,
\tag{2}
$$

with flux through the surface of the sphere given by $\int dA = 4\pi s$.[2] Because of the background flux, the lowest energy states are the LLL states. Restricting to the LLL contains $2s + 1$ degenerate orbitals for each spin, in which the fermions can be expanded as

$$
\psi_i(\Omega) = \frac{1}{R} \sum_{m=-s}^{s} \Phi_m(\Omega) c_{m,i} \qquad (i = \uparrow, \downarrow),
\tag{3}
$$

with

$$
\Phi_m(\Omega) = N_m e^{im\phi} \cos^{s+m}\left(\frac{\theta}{2}\right) \sin^{s-m}\left(\frac{\theta}{2}\right), \qquad N_m^2 = \frac{(2s+1)!}{4\pi(s+m)!(s-m)!}.
\tag{4}
$$

Each state at half-filling has $N = 2s + 1$ fermions.

The restriction to the LLL is a UV regulator, which implements a rotationally invariant UV cutoff on length scales shorter than $\Lambda_{\rm UV}^{-1} \sim |B|^{-1/2}$, where $|B|$ is the background magnetic field. One way to see this is to look at how the completeness relation for the sum over modes is modified by discarding the higher LLs:

$$
\{\psi_i^\dagger(\Omega), \psi_i(\Omega')\} = \frac{1}{R^2} \sum_{m=-s}^{s} \Phi_m^*(\Omega)\Phi_m(\Omega') = \frac{2s+1}{4\pi R^2} \cos^{2s}\frac{\delta\theta}{2},
\tag{5}
$$

where $\Omega \cdot \Omega' \equiv \cos\delta\theta$. At large $s$, (5) approaches a $\delta$ function smeared out over angles, $\delta\theta \sim 1/\sqrt{s}$, or equivalently over lengths $\delta x = R\delta\theta \sim |B|^{-1/2}$. It is convenient to use units set by this UV scale, and in particular we will take $|B| \equiv 1/2$ so that $R^2 = N - 1$.

---

[2] We have used differential forms on $S^2$ to express $A$, but one also commonly sees these formulas for the monopole in vector calculus notation in $\mathbb{R}^3$, as follows. The unit vector $\hat{e}_\phi \equiv (-\cos\theta, \sin\theta, 0)$ is related to $d\phi$ by $d\phi \cong \sqrt{g^{\phi\phi}}\hat{e}_\phi = \frac{1}{R\sin\theta}\hat{e}_\phi$, and then the vector $A$ can be written $\vec{A} = -\hat{e}_\phi \frac{s}{R}\cot\theta$. The magnetic field is $\vec{B} = \nabla \times \vec{A} = \frac{s}{R^2}\hat{r}$, and the flux is $R^2 \int d^2\Omega \hat{r} \cdot \vec{B} = \int dA$.

Finally, the Hamiltonian restricted to the LLL states follows from the action and is the integral $H = R^2 \int d^2\Omega \mathcal{H}$ over the Hamiltonian density $\mathcal{H}$:

$$\mathcal{H} = \sum_n \left( \lambda_n (\psi^\dagger \psi) \frac{\nabla^{2n}_{S^2_1}}{R^{2n}} (\psi^\dagger \psi) - \lambda_{n,z} (\psi^\dagger \sigma^z \psi) \frac{\nabla^{2n}_{S^2_1}}{R^{2n}} (\psi^\dagger \sigma^z \psi) \right) - h\psi^\dagger \sigma^x \psi \,, \tag{6}$$

plus an implicit vacuum energy. The operators are not normal ordered. The factors of $R^{2n}$ in the denominator come from writing the Laplacian in terms of the Laplacian $\nabla^2_{S^2_1}$ on the unit sphere, $\vec{\nabla}^2 = \nabla^2_{S^2_1}/R^2$. From now on, we will work only with $\nabla^2_{S^2_1}$ and so will drop the subscript. We will also follow [6] and restrict to the case $\lambda_{n,z} = \lambda_n$.[3]

## 3  Constructing the conformal algebra generators

### 3.1  Generators from $T^{\mu\nu}$

In a conformal theory, all the Noether currents $j^\mu_\epsilon$ for the conformal symmetries $x^\mu \to x^\mu + \epsilon^\mu(x)$ can be written in terms of the energy-momentum tensor,

$$j^\mu_\epsilon(x) = \epsilon^\nu(x) T^\mu_{\ \nu}(x), \tag{7}$$

and so the corresponding generators $Q_\epsilon$ of the transformations are all spatial integrals over $T^{00}$ and $T^{0a}$ of the form $Q_\epsilon = \int d^{d-1}x \sqrt{g}\, j^0_\epsilon(x)$.[4] We review how to derive the conformal generators on $\mathbb{R} \times S^2$ in terms of $T^{\mu\nu}$ in appendix A. The Dilatation generator is just (proportional to) the Hamiltonian and depends only on $T^0_{\ 0}$:

$$D = \int d^2\Omega\, T^0_{\ 0}. \tag{8}$$

The rotation generators $J_z, J_\pm$ are given by the following integrals of $T^0_{\ a}$:

$$J_z \propto \int d^2\Omega T^0_{\ \phi}, \qquad J_\pm \propto \pm i \int d^2\Omega e^{\pm i\phi} (T^0_{\ \theta} \pm i\cot\theta\, T^0_{\ \phi}), \tag{9}$$

and can be written in embedding space notation (see appendix A) for $S^2 \subset \mathbb{R}^3$ as

$$J^B \propto \int d^2\Omega \epsilon^{ABC} \widehat{x}^C T^{0A}. \tag{10}$$

The generators $P_A$ of translations and $K_A$ of special conformal transformations (SCTs) can be written in embedding space notation as

$$P^A = \int d^2\Omega(\widehat{x}^A T^0_{\ 0} + iT^{0A}), \qquad K^A = \int d^2\Omega(\widehat{x}^A T^0_{\ 0} - iT^{0A}). \tag{11}$$

Note that $P+K$ depends only on $T^0_{\ 0}$ and $P-K$ depends only on $T^{0A}$, and $P^\dagger_A = K_A$.

---

[3] The combination $(\psi^\dagger \psi)^2 + (\psi^\dagger \sigma^z \psi)^2$ is proportional to the identity on the LLL, so changing $\lambda_0 \to \lambda_0 + c, \lambda_{0,z} \to \lambda_{0,z} - c$ has no effect on the theory, and one can set $\lambda_0 = \lambda_{0,z}$ without loss of generality.

[4] In the $3d$ Ising model, there is no ambiguity about which operator is the correct energy-momentum tensor, because it is the unique dimension-3 spin-2 operator; in a sense, this is the 'generic' case, since the presence of multiple such operators requires additional symmetries.

## 3.2 Conformal generators from $\mathcal{H}$

### 3.2.1 $\mathcal{H}$ vs $T^0_{\ 0}$

If we had direct access to the energy momentum tensor of the CFT, the above expressions would be all we would need to construct the conformal generators. In practice, however, the fuzzy sphere construction is only conformal in the infrared (IR), and the emergent stress tensor in the CFT does not correspond to a simple local operator in the microscopic description. Instead, there is a nontrivial mapping between UV and IR operators, and a local operator in the UV will generically be represented as an infinite sum over all local operators in the CFT allowed by symmetry. Consequently, a conceptually straightforward strategy for obtaining the CFT stress tensor is to take the microscopic Hamiltonian density $\mathcal{H}$ and analyze its expansion in a basis of CFT operators, e.g.

$$\mathcal{H} = \gamma T^0_{\ 0} + \sum_{\mathcal{O}} g_{\mathcal{O}} \mathcal{O}. \tag{12}$$

By symmetry, the operators $\mathcal{O}$ in this expansion should be scalars under $SO(3)$ rotations, including for instance scalar components of spinning operators such as $T^{00}$.

Aside from being a fairly natural choice, starting the construction of $T^0_{\ 0}$ with the Hamiltonian density $\mathcal{H}$ has some practical advantages. The first is that the dilatation operator is (proportional to) the Hamiltonian on $\mathbb{R} \times S^2$, and so

$$H = R^2 \int d^2\Omega \, \mathcal{H}, \tag{13}$$

is the exact generator (by definition) of time translations, even away from the critical point. A second, related, advantage of using $\mathcal{H}$ is that it is straightforward to show that at linear order, any operators with time derivatives in the expansion above can be removed by a basis rotation.[5]

Moreover, because the Hamiltonian is the integral over $\mathcal{H}$, the same expansion of $\mathcal{H}$ above shows up for the expansion of the Hamiltonian around the CFT limit:

$$H = \gamma H_{\text{CFT}} + \sum_{\mathcal{O} \text{ primary}} g_{\mathcal{O}} \int d^2\Omega \, \mathcal{O}(\Omega). \tag{14}$$

Any descendant operators in the expansion are total spatial derivatives and vanish by integration by parts, so only primaries survive. The fact that these coefficients $g_{\mathcal{O}}$ for primary operators in the Hamiltonian are the same as in the expansion of $\mathcal{H}$ is useful for two practical reasons. First, it means that slightly away from the critical point, the values of the $g_{\mathcal{O}}$s can be inferred by inspection of the spectrum of eigenvalues of $H$ and comparison with conformal perturbation theory. Second, it means that we can actually set a finite number of the $g_{\mathcal{O}}$s to zero simply by tuning the parameters in the microscopic theory to bring us closer to the critical point.[6] Consequently, the main new feature that arises in the expansion of $\mathcal{H}$ is that one must also consider the effect of descendant operators, which have no effect on the spectrum of energies but do affect the conformal generators.

As reviewed in section 2, at the critical point there are two intrinsic physical energy scales in the system we study, which we can call the UV scale or "lattice" scale $\Lambda_{\text{UV}}$ (though the fuzzy

---

[5]The argument is essentially the one given in [19] for time derivatives in the Hamiltonian itself. For any operator $\mathcal{O}$, consider the basis rotation $U \equiv e^{i\lambda \int d^2\hat{y} \mathcal{O}(\hat{y})}$. Under this basis change, $\mathcal{H}(\hat{x}) \to U\mathcal{H}(\hat{x})U^\dagger = \mathcal{H}(\hat{x}) + i\lambda \int d^2\hat{y}[\mathcal{O}(\hat{y}), \mathcal{H}(\hat{x})] + \mathcal{O}(\lambda^2) = \mathcal{H}(\hat{x}) + \lambda\frac{d}{dt}\mathcal{O}(\hat{x}) + \mathcal{O}(\lambda^2)$.

[6]The details of this tuning will be described in detail in subsequent sections, but the idea is that at any finite size of the fuzzy sphere, there are an infinite number of irrelevant operators in the expansion of $H$ around $H_{\text{CFT}}$, and their presence can be detected by comparing the spectrum of $H$ to the spectrum of $H_{\text{CFT}}$ when the latter is known.

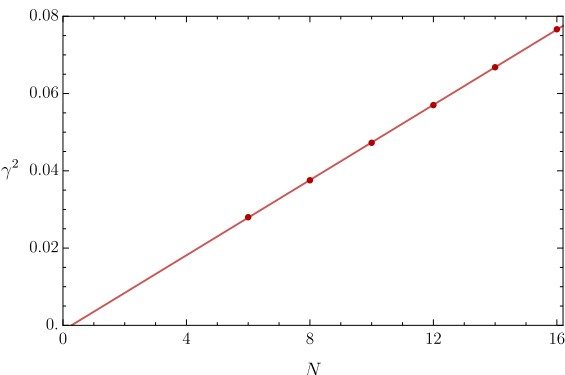

Figure 1: Dependence of $\gamma^2 \propto \Lambda_{\mathrm{UV}}^2/\Lambda_{\mathrm{IR}}^2$ on $N$; to good approximation, $\Lambda_{\mathrm{UV}}^2/\Lambda_{\mathrm{IR}}^2 \propto N - 0.25$.

sphere does not have a lattice, it is still convenient to use the language of lattice regularizations) and the IR scale $\Lambda_{\mathrm{IR}}$, which we define as the size of the energy gap in the spectrum and which is inversely proportional to the radius $R$ of the sphere. Restoring units of $\Lambda_{\mathrm{UV}}$,

$$\mathcal{H} = \sum_{\mathcal{O}} \widehat{g}_{\mathcal{O}} \Lambda_{\mathrm{UV}}^{3-\Delta_{\mathcal{O}}} \mathcal{O}, \tag{15}$$

where $\widehat{g}_{\mathcal{O}}$ is dimensionless and set by the microscopic theory, and therefore independent of the size of $N$. However, as $N$ increases, the sphere increases in size, and consequently the gap (at the critical coupling) $\Lambda_{\mathrm{IR}} \sim 1/R$ will decrease, so that

$$\Lambda_{\mathrm{IR}}^2/\Lambda_{\mathrm{UV}}^2 \sim 1/N. \tag{16}$$

At each value of $N$, one rescales energies to get dimensions by fixing the dimension of the stress tensor to be $\Delta_T = \gamma E_T \equiv 3$, so $\gamma = 3/E_T \sim 1/\Lambda_{\mathrm{IR}}$ and $\gamma^2$ is proportional to $\Lambda_{\mathrm{UV}}^2/\Lambda_{\mathrm{IR}}^2$. In Fig. 1, we show the numerical dependence of $\gamma^2$ on $N$, where one sees that to very good approximation $\Lambda_{\mathrm{UV}}^2/\Lambda_{\mathrm{IR}}^2 \approx N - 0.25$.[7] After rescaling all dimensional quantities by $\gamma$, the expansion of $\mathcal{H}$ takes the form

$$\mathcal{H} = \sum_{\mathcal{O}} \widehat{g}_{\mathcal{O}} \left(\frac{\Lambda_{\mathrm{UV}}}{\Lambda_{IR}}\right)^{3-\Delta_{\mathcal{O}}} \mathcal{O} \sim \sum_{\mathcal{O}} \widehat{g}_{\mathcal{O}} N^{\frac{3-\Delta_{\mathcal{O}}}{2}} \mathcal{O}. \tag{17}$$

### 3.2.2 Recipe for constructing generators

From the expressions (11), it might seem most natural to construct $P^A$ and $K^A$ separately by constructing $T^{00}$ and $T^{0A}$. However, we will take a different approach and first construct the combination

$$\Lambda^A \equiv P^A + K^A = 2 \int d^2\Omega \, \widehat{x}^A T^0_{\ 0}, \tag{18}$$

since this combination depends only on $T^0_{\ 0}$. Because Lorentz invariance is broken along the RG flow, a priori we do not have any simple relation between the representation of $T^{00}$ and $T^{0A}$ in terms of microscopic operators. One might hope that since rotations are exactly preserved by the fuzzy sphere regulator, that might provide a useful handle on $T^{0A}$ as a local operator built from the Noether charge density $V_A$ for rotations. Unfortunately, as we discuss in appendix B, $V_A$ does not appear to be useful for constructing $P^A - K^A$.

---

[7]See also [6] Fig. 7 for an equivalent result.

However, there is a more practical approach that we can take for obtaining $P^A$ and $K^A$ once we have the combination $\Lambda_A \equiv P_A + K_A$. One way to see that it is not necessary to independently construct $P - K$ is that the entire conformal algebra is generated by $D, J_A$ and $P_z + K_z$. So, one could obtain $P - K$ from the relation

$$P - K = [D, P + K]. \tag{19}$$

In fact, the approach we will follow is even easier in practice. The key point is that in the CFT, $P$ acting on a state raises its dimension by 1 and $K$ acting on a state lowers it by 1. Since we are numerically diagonalizing $D$, the dimension of all states is known, and so one can separate out the contributions to $P + K$ coming from $P$ versus those coming from $K$ by looking at the difference in dimension between the bra and ket state. In Fig. 2, we plot the size of $|(P_z + K_z)_{ij}|$ against the difference in dimension $\Delta_{ij} \equiv \Delta_i - \Delta_j$ for all eigenstates up to dimension 5.5. There are clearly two large spikes, around $\Delta_{ij} = \pm 1$, with the size of the matrix elements decreasing away from these points. In fact, the extent to which acting with $P$ or $K$ on a state mostly produces only states with $\Delta \to \Delta \pm 1$ provides a useful quantitative measure of the validity of the CFT generators being constructed. The construction we will use in this paper is that $P_A$ includes all the matrix elements of $\Lambda_A$ that raise the dimension and $K_A$ includes all the matrix elements of $\Lambda_A$ that lower the dimension.[8] One could easily use a different construction by imposing different restrictions on the change in dimensions for the $P_A$ and $K_A$ components, but this definition seems natural to us in that it preserves the fact that $P_A + K_A = \Lambda_A$.

In appendix C, we provide expressions for constructing $\Lambda_A$ from the fuzzy sphere Hamiltonian density. In practice, we will only ever compute the matrix elements of $\Lambda_z$, and moreover we will only compute them in the $j_z = 0$ and $j_z = 1$ sectors. By the Wigner-Eckart theorem, we can obtain all other matrix elements of $\Lambda_A$ between all other states from this subset alone.

## 4 Matching and tuning the generators

### 4.1 Conformal perturbation theory for $K + P$

Our main goal in this section will be to see how to remove as much as possible the corrections to the generators for $(P + K)_A$ in order to obtain their values in the conformal limit. Because corrections to $T_0^0$ coming from primary operators affect the Hamiltonian whereas corrections from descendant operators do not, the role of primaries and descendants will be qualitatively different. Primary operators contribute both directly to the matrix elements of $(P + K)$ through their explicit presence in the operator, as well as through the fact that they modify the Hamiltonian and therefore affect the eigenstates. Our strategy will be to try tuning them away directly at the level of the microscopic Lagrangian as much as possible by analyzing the spectrum of eigenvalues. Then the dominant corrections to the matrix elements of $(P + K)$ will all come from descendant operators, making the analysis cleaner and simpler.

In the CFT limit, the matrix elements of $P_z$ and $K_z$ are nonzero only between states within a single representation, by definition. Given a primary operator $\mathcal{O}$, with a corresponding primary state $|\mathcal{O}\rangle$, the 'level-$n$' descendants are all states obtained from $|\mathcal{O}\rangle$ by acting with $n$ factors of $P_A$.

Consider first for simplicity the case where the primary $\mathcal{O}$ is a scalar operator. Then in $3d$, at each level $n$ there is a unique state with spin $\ell$ for allowed values of spin, $0 \le \ell \le n, n - \ell = 0$ (mod 2). Let $(n, \ell)$ denote the descendant at level $n$ with spin $\ell$. The matrix elements of $P_z$ are

---

[8]Since the operator $\Lambda_A$ carries spin 1, its diagonal matrix elements vanish by rotational invariance.

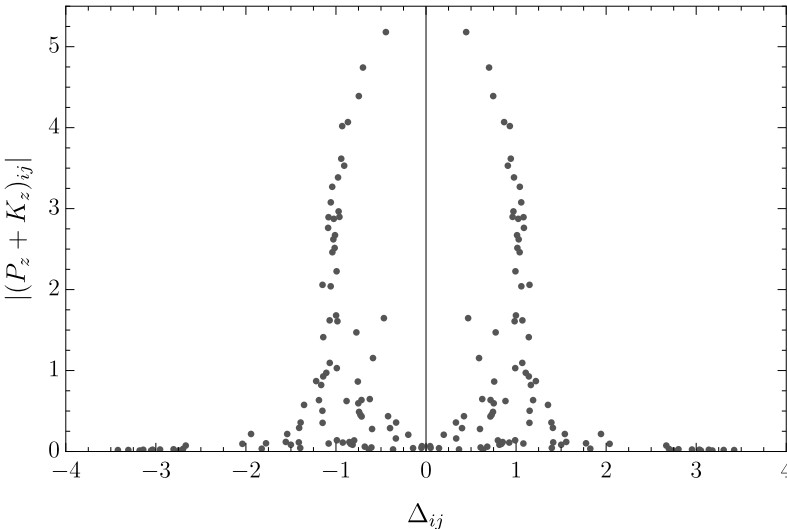

Figure 2: Plot of the size of $|(P_z + K_z)_{ij}|$ versus the difference in dimension $\Delta_{ij} \equiv \Delta_i - \Delta_j$ for all eigenstates up to dimension 5.75, using $N = 16$, and setting $j_z = 0$ but otherwise including all symmetry sectors. Matrix elements less than $10^{-2}$ are not shown. There are clearly two large features centered around $\Delta_{ij} = \pm 1$, indicating that for the vast majority of nonnegligible matrix elements looking at $\Delta_{ij}$ provides a simple way to separate out the contributions of $P$ vs $K$ to the sum $P + K$. Parameters were $V_0 = 4.825, h = 3.158$, and we have used three parameters to tune away derivative contributions inside $T_0^0$ as described in section 4.

nonzero only between $(n, \ell)$ and $(n+1, \ell+1)$ or between $(n, \ell)$ and $(n+1, \ell-1)$; in appendix E we show that

$$\langle n+1, \ell'|P_z|n, \ell\rangle_{\text{CFT}}^2 = \begin{cases} \frac{\ell^2(n-\ell+2)(2\Delta+n-\ell-1)}{(2\ell+1)(2\ell-1)} & (\ell' = \ell - 1), \\ \frac{(\ell+1)^2(n+\ell+3)(2\Delta+n+\ell)}{(2\ell+1)(2\ell+3)} & (\ell' = \ell + 1), \end{cases} \tag{20}$$

where $\Delta$ is the dimension of the scalar primary state of the representation. For the case of primaries with general spin, appendix E additionally contains an efficient recursion relation for obtaining the matrix elements of $P_A$. The results for several low-lying states are depicted in Fig. 3.

In practice, it is not generally possible to tune away all deviations from the conformal limit. Even at the critical coupling, there will be irrelevant interactions that scale to zero only in the infinite IR, which is not possible to access numerically. Instead, as in (12),

$$\mathcal{H} = \gamma T_0^0 + \sum_{\mathcal{O}} g_{\mathcal{O}} \mathcal{O},$$

$$H = \gamma H_{\text{CFT}} + V, \qquad V \equiv \sum_{\mathcal{O}} g_{\mathcal{O}} \int d^2\Omega \, \mathcal{O}, \tag{21}$$

where $\mathcal{H}$ and $H$ are the Hamiltonian density and Hamiltonian in the deformed theory. When we diagonalize $H$, if $V$ is sufficiently small, then each eigenstate $|\widetilde{\mathcal{O}}_n\rangle$ can be identified as a CFT state $|\mathcal{O}_n\rangle$ plus corrections. At linear order in the deformation $V$,

$$|\widetilde{\mathcal{O}}_n\rangle = |\mathcal{O}_n\rangle + \sum_m \frac{\langle \mathcal{O}_m|V|\mathcal{O}_n\rangle}{\Delta_n - \Delta_m}|\mathcal{O}_m\rangle. \tag{22}$$

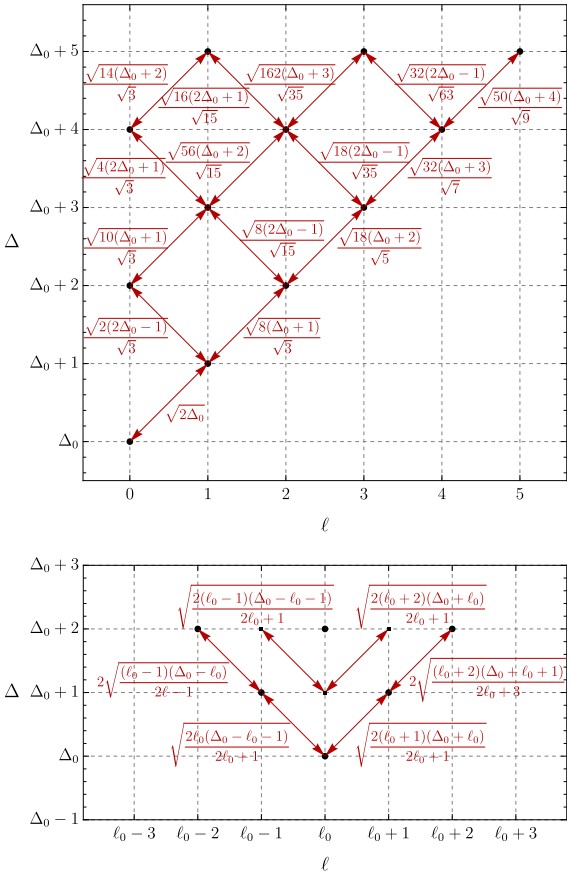

Figure 3: CFT predictions for the matrix elements of $\Lambda_z$ for some low-lying states, for a spin-0 primary (top) and spin-$\ell_0$ primary (bottom) of dimension $\Delta_0$. Parity-even (odd) states are depicted with circles (squares). When $\ell_0 > 0$, there are two states at dimension $\Delta = \Delta_0 + 2$ and spin $\ell = \ell_0$.

The sum over $m$ here is over all states, including descendants.

The matrix elements of $\mathcal{H}$ differ from those of $T^0_{\ 0}$ in the CFT because of two types of effects. The first comes directly from the difference $\mathcal{H} - T^0_{\ 0}$ as an operator. The second difference is indirect, due to the difference $|\widetilde{\mathcal{O}}_n\rangle - |\mathcal{O}_n\rangle$ between the eigenstates in the CFT and in the deformed theory. We can separate out these two effects as follows:

$$\widetilde{F}_{\mathcal{O}_r T \mathcal{O}_n}(x) \equiv \langle \widetilde{\mathcal{O}}_r | \mathcal{H}(x) | \widetilde{\mathcal{O}}_n \rangle = \widetilde{F}^{(\text{op})}_{\mathcal{O}_r T \mathcal{O}_n}(x) + \widetilde{F}^{(\text{state})}_{\mathcal{O}_r T \mathcal{O}_n}(x),$$

$$\widetilde{F}^{(\text{op})}_{\mathcal{O}_r T \mathcal{O}_n}(x) \equiv \gamma \langle \mathcal{O}_r | T^{00}(x) | \mathcal{O}_n \rangle + \sum_{\mathcal{O}} g_{\mathcal{O}} \langle \mathcal{O}_r | \mathcal{O}(x) | \mathcal{O}_n \rangle, \tag{23}$$

$$\widetilde{F}^{(\text{state})}_{\mathcal{O}_r T \mathcal{O}_n}(x) \equiv \sum_{\mathcal{O} \text{ primary}} g_{\mathcal{O}} \sum_m \left[ \frac{\langle \mathcal{O}_r | T^{00}(x) | \mathcal{O}_m \rangle \langle \mathcal{O}_m | \int d^2 y \sqrt{g} \mathcal{O}(y) | \mathcal{O}_n \rangle}{\Delta_n - \Delta_m} \right.$$

$$\left. + \frac{\langle \mathcal{O}_r | \int d^2 y \sqrt{g} \mathcal{O}(y) | \mathcal{O}_m \rangle \langle \mathcal{O}_m | T^{00}(x) | \mathcal{O}_n \rangle}{\Delta_r - \Delta_m} \right].$$

By inspection, these sums can be rewritten in the form of integrals over time-ordered correlators. The sum over $m$ is just a sum over states with an energy denominator, which can be written as a sum over states with an integral over time, with time-ordered operators. Assuming

that $\Delta_m > \Delta_n, \Delta_r$,[9] the integrals converge and one finds

$$
\begin{aligned}
\widetilde{F}^{(\text{state})}_{\mathcal{O}_r T \mathcal{O}_n}(x) &= \sum_{\mathcal{O} \text{ primary}} g_{\mathcal{O}} \int_{-\infty(1-i\epsilon)}^{\infty(1-i\epsilon)} (-i) dt \int d^2 y \sqrt{g} \langle \mathcal{O}_r | T\{\mathcal{O}(t,y) T^{00}(0,x)\} | \mathcal{O}_n \rangle \\
&= -\sum_{\mathcal{O} \text{ primary}} g_{\mathcal{O}} \int_{-\infty}^{\infty} dt_E \int d^2 y \sqrt{g} \langle \mathcal{O}_r | \mathcal{O}(-it_E, y) T^{00}(0,x) | \mathcal{O}_n \rangle ,
\end{aligned}
\tag{24}
$$

where in the second line we have Wick rotated to obtain an integral over Euclidean time $t_E = it$. In general, evaluating $\widetilde{F}^{(\text{state})}_{\mathcal{O}_r T \mathcal{O}_n}(x)$ requires knowledge of the four-point function that appears above. However, we will usually restrict to the main case of interest where $|\mathcal{O}_n\rangle$ and $|\mathcal{O}_r\rangle$ have spins $\ell_n$ and $\ell_r$ that differ by 1 ($|\ell_n - \ell_r| = 1$). In this case, by rotational invariance the only part of $T^{00}(0,x)$ that contributes is its integral against $\widehat{x}$, which therefore reduces it to $\Lambda_z = K_z + P_z$, and the correlator reduces to a three-point function. If $|\mathcal{O}_r\rangle$ or $|\mathcal{O}_n\rangle$ is the vacuum, then the correlator reduces even further, to a two-point function.

In appendix D, we work out $F_{\mathcal{O}_r T \mathcal{O}_n}$ in detail for the case where $\mathcal{O}_r = 1$ is the identity and $\mathcal{O}_n = \mathcal{O}_{1,1}$ is a level-1, spin-1 descendant of a scalar primary $\mathcal{O}$. In this case, the result is

$$
\widetilde{F}_{1 T \mathcal{O}_{1,1}}(\widehat{x}) = \cos\theta \left( g_{\mathcal{O}} \sqrt{\frac{2}{3\Delta}} \frac{\Delta - 3}{\Delta + 2} + \sum_{n=1}^{\infty} (-2)^n g_{\nabla^{2n}\mathcal{O}} \sqrt{\frac{2\Delta}{3}} \right),
\tag{25}
$$

where $\Delta = \Delta_{\mathcal{O}}$. If $\mathcal{O}$ is allowed by symmetry to appear in $\mathcal{H}$, then so are all of its scalar descendants, and generically one would expect all of them to be present. By inspection of this formula, their contributions cannot be distinguished from each other based on their $\widehat{x}$ dependence (which is also clear from the symmetries of the bra and ket states). On the other hand, the coefficients $g_{\nabla^{2n}\mathcal{O}}$ all have different scaling dimensions, and they *can* be distinguished from each other based on their dependence on the size of the fuzzy sphere, i.e. on $\Lambda_{\text{UV}}/\Lambda_{\text{IR}}$.

## 4.2 Matching and tuning the spectrum

As discussed in Sec. 3.2.1, we can think about the UV Hamiltonian as an expansion around the CFT one plus contributions from $\mathbb{Z}_2$-even primaries. To define our critical point we can therefore think of tuning the $\lambda_n$ and $h$ parameters in $H$ to minimize the corrections to the spectrum coming from relevant and slightly irrelevant operators. At first order in perturbation theory

$$
H = \gamma H_{\text{CFT}} + \sum_{\mathcal{O}} g_{\mathcal{O}} \int d^2\Omega \, \mathcal{O}(\Omega), \qquad H |\mathcal{O}_i\rangle = E_i |\mathcal{O}_i\rangle ,
$$

$$
E_i = \gamma \Delta_i + \delta E_i^{(\mathcal{O})}, \qquad\qquad \delta E_i^{(\mathcal{O})} = \frac{\langle \mathcal{O}_i | \sum_{\mathcal{O}} g_{\mathcal{O}} \int d^2\Omega \, \mathcal{O}(\Omega) | \mathcal{O}_i \rangle}{\langle \mathcal{O}_i | \mathcal{O}_i \rangle} .
\tag{26}
$$

The expression for $\delta E_i$ depends on both the spin of the external state and whether it is a primary or a descendant. If we focus on cases where $\mathcal{O}_i$ is either scalar primary or its first descendant state, the corresponding expressions for $\delta E_i$ [19] are:

$$
\delta E_i^{(\mathcal{O})} = \begin{cases} g_{\mathcal{O}} f_{\widetilde{\mathcal{O}}_i \mathcal{O} \widetilde{\mathcal{O}}_i}, & \mathcal{O}_i = \widetilde{\mathcal{O}}_i \text{ primary}, \\ g_{\mathcal{O}} f_{\widetilde{\mathcal{O}}_i \mathcal{O} \widetilde{\mathcal{O}}_i} \left( 1 + \frac{\Delta_{\mathcal{O}}(\Delta_{\mathcal{O}} - 3)}{6\Delta_{\widetilde{\mathcal{O}}_i}} \right), & \mathcal{O}_i = \partial \widetilde{\mathcal{O}}_i, \end{cases}
\tag{27}
$$

with $f_{\widetilde{\mathcal{O}}_i \mathcal{O} \widetilde{\mathcal{O}}_i}$ the OPE coefficient.

---

[9]More generally, a finite number of terms with $\Delta_m < \Delta_n, \Delta_r$ may be separated out and dealt with independently.

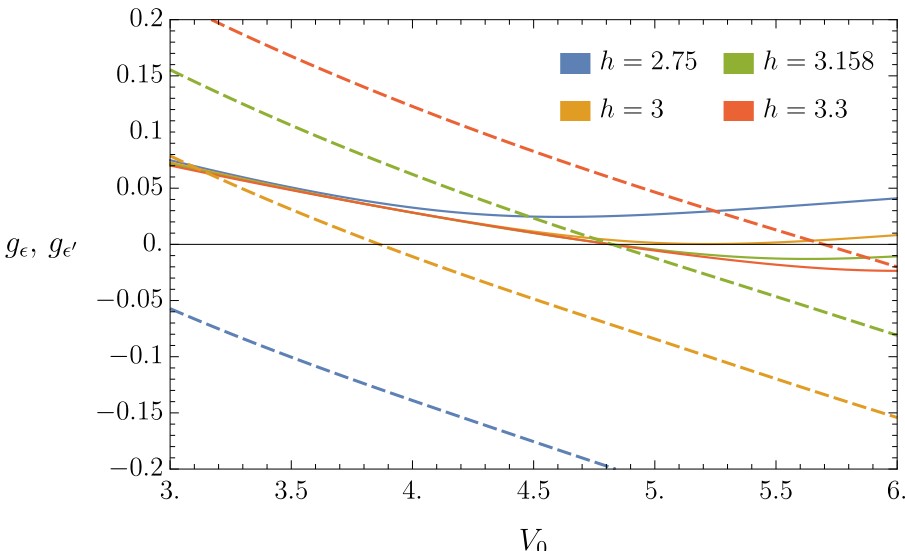

Figure 4: Plot of extracted values for the Wilson coefficients $g_\epsilon$ (dashed) and $g_{\epsilon'}$ (solid) of the operators $\epsilon$ and $\epsilon'$ in the Hamiltonian, as a function of $V_0$ and $h$ in the microscopic theory at $N = 12$. One can see that both $g_\epsilon$ and $g_{\epsilon'}$ vanish at approximately $V_0 = 4.825, h = 3.158$.

In practice, in our analysis we consider corrections coming from $\epsilon$ and $\epsilon'$, the lowest dimension $\mathbb{Z}_2$-even scalar primary.[10] Focusing on a subset of operators, $\mathcal{O}_i = \epsilon, \sigma, \epsilon', \partial\epsilon\,\partial\sigma$, we determine $g_\epsilon$, $g_{\epsilon'}$, $\gamma$ by minimizing

$$\min_{\gamma, g_\epsilon, g_{\epsilon'}} \sum_{\mathcal{O}_i} \left( E_i - \gamma\Delta_i - \delta E_i^{(\epsilon)} - \delta E_i^{(\epsilon')} \right)^2 . \tag{28}$$

In Fig. 4 we show the Wilson coefficients $g_\epsilon$ and $g_{\epsilon'}$, extracted at $N = 12$, as a function of $h$ and the Haldane pseudopotentials $V_0, V_1 = 1$, which are related to the microscopic parameters $\lambda_n$ as (see appendix C for more details)

$$
\begin{aligned}
\frac{\lambda_0}{R^2} &= \frac{2\pi}{(2s+1)^2} \left( (4s+1)V_0 + (4s-1)V_1 \right), \\
\frac{\lambda_1}{R^4} &= \frac{2\pi}{(2s+1)^2} \cdot \frac{(4s-1)V_1}{s}.
\end{aligned}
\tag{29}
$$

We then define the critical point as the values of $V_0$ and $h$ such that

$$g_\epsilon = 0 = g_{\epsilon'} \quad \Longleftrightarrow \quad V_0 = 4.825, \quad h = 3.158 \quad (\text{and } V_1 = 1). \tag{30}$$

In Fig. 5, we compare this point to the original choice of parameters in [6] and the one minimizing the errors between the dimensions obtained and known conformal bootstrap results for some low dimensional operators. Notice how both our choice of parameters and the one in [6] lie inside the regions minimizing the errors and therefore they are equivalently good at the level of the spectrum. The reason why we opted to define the critical point as (30) is that this method is a more direct attempt to isolate and remove the most relevant deformations to the Hamiltonian, which is what one would want to do in order to obtain the fastest convergence with $N$.

---

[10]See appendix F for all the necessary OPE data.

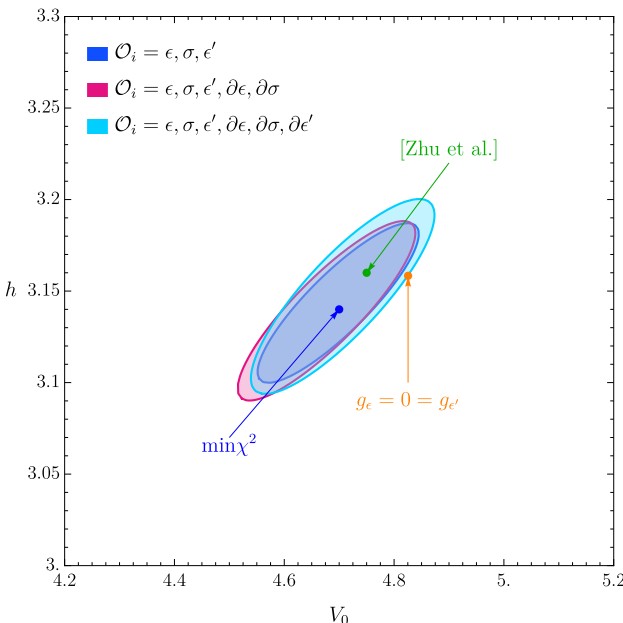

Figure 5: Plot of the $\chi^2$ for the comparison of fuzzy sphere dimensions versus bootstrap data, where $\chi^2 \equiv \frac{100}{N_{\text{ops}}} \sum_{i=1}^{N_{\text{ops}}} (\Delta_i - \Delta_i^{(\text{CB})})^2 / \Delta_i^{(\text{CB})}$. Various choices are depicted for the states included in the $\chi^2$; in all cases, the contours are $\chi^2 = 0.01$. Also shown is the point where $g_\epsilon = g_{\epsilon'} = 0$ from Fig. 4.

## 4.3 Tuning the generators

Next, we will turn to the issue of tuning away descendant operators in $\mathcal{H} - T_0^0$. In Fig. 6, we show the result for the matrix element $\langle \text{vac} | \widetilde{\Lambda}_z | \partial \epsilon \rangle$ of $\widetilde{\Lambda}_z$ between the vacuum and the spin-1, level-1 descendant state $|\partial \epsilon\rangle$, and the analogous plot for $\langle \text{vac} | \widetilde{\Lambda}_z | \partial \epsilon' \rangle$, computed in the fuzzy sphere at $V_0 = 4.825, h = 3.158$ for $N = 6, 8, \ldots, 16$. We use a tilde on $\Lambda$ to denote the matrix elements computed at finite $N$, which will be contaminated by deviations from the CFT limit. The expected CFT result is that these particular matrix elements should vanish in the IR, at $N = \infty$. Interestingly, up to $N = 16$, the highest we consider, the result for $\langle \text{vac} | \widetilde{\Lambda}_z | \partial \epsilon \rangle$ appears to be getting worse as $N$ increases. However, if one fits all the data as a function of $N$ to a power series with the expected powers based on the dimension of $\epsilon$, the fit is fairly stable and correctly predicts that the result at $N = \infty$ should vanish (to within numeric error). The coefficients in the fit for the $\partial \epsilon'$ case are larger than those for $\partial \epsilon$, which is expected based on the higher dimension $\Delta_{\partial \epsilon'} / \Delta_{\partial \epsilon} \approx 1.9$, so the IR scale $\Lambda_{\text{IR}}$ is correspondingly higher and therefore the expansion parameter $\left( \frac{\Lambda_{\text{IR}}}{\Lambda_{\text{UV}}} \right)^2 \propto \frac{\Lambda_{\text{IR}}^2}{N}$ is roughly a factor of 4 larger.

The main takeaway from Fig. 6 is that convergence with $N$ is extremely slow even if one tunes to the critical coupling for $V_0$ and $h$. This slow convergence is due to the descendant operator $\nabla^2 \epsilon$, which has dimension $\sim 3.41$ and therefore converges with $N$ like $N^{-0.206}$. In the fits, this contribution shows up as a 'nosedive' to the correct value at $1/N = 0$, and to suppress it by taking large $N$ would require taking impractically large values of $N$. Instead, it will be vastly more efficient to correct $\widetilde{\Lambda}_A$ by adding additional local microscopic operators to $\mathcal{H}$ and tuning their coefficients to remove the contamination from $\nabla^2 \epsilon$. Fortunately, we can easily do better than removing $\nabla^2 \epsilon$. At a fixed $N$, the contributions to $\widetilde{\Lambda}$ from descendant operators of the same primary are indistinguishable from each other, and can be removed in one fell swoop. To make this explicit, write out the expansion of $\int d^2 \Omega \, x^A \mathcal{H}$ in terms of CFT

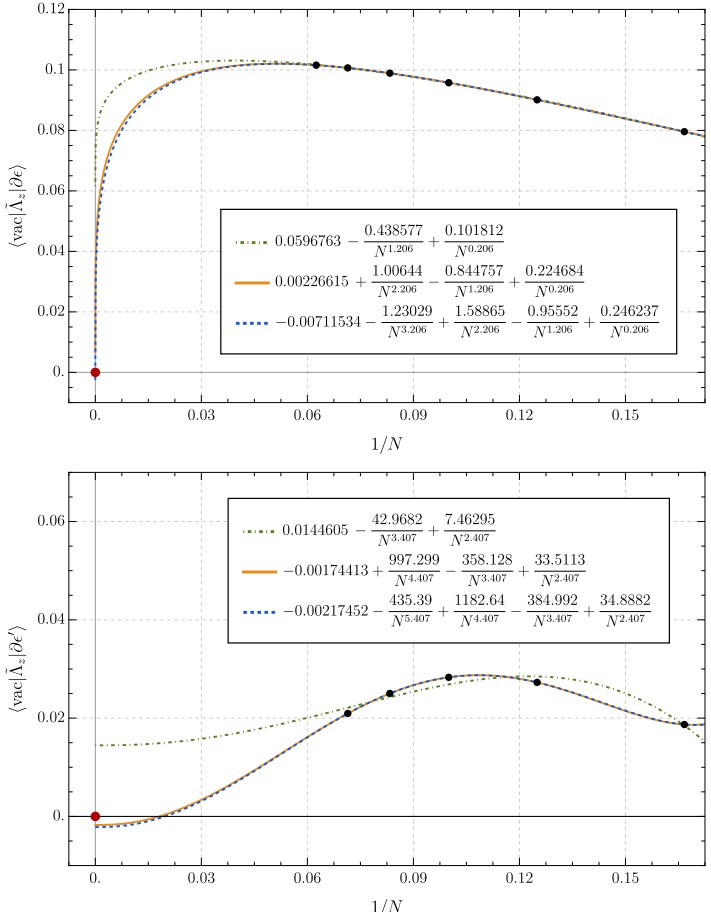

Figure 6: Matrix element of $\widetilde{\Lambda}_z$ between the vacuum and the spin-1, level-1 descendant state $|\partial\epsilon\rangle$ (top) and $|\partial\epsilon'\rangle$ (bottom), computed in the fuzzy sphere at $V_0 = 4.825, h = 3.158$ for $N = 6, 8, \ldots, 16$. We also show fits to the data, where the powers in the Taylor series are those expected based on the dimensions of $\epsilon, \epsilon'$ and its descendants. The expected result in the CFT is 0, shown as a red dot, which is not included as an input to the fit but rather is an output used to test the reliability of the fit.

operators, and focus on operators in the $\mathcal{O}$ representation:

$$\widetilde{\Lambda}_A \equiv 2 \int d^2\Omega \, x_A \mathcal{H} \supset 2 \int d^2\Omega \, x_A \sum_{n=0}^{\infty} g_{\nabla^{2n}\mathcal{O}} \nabla^{2n}\mathcal{O} = 2 \left[ \sum_{n=0}^{\infty} (-2)^n g_{\nabla^{2n}\mathcal{O}} \right] \int d^2\Omega \, x_A \mathcal{O}, \quad (31)$$

where we have integrated by parts and used the fact that $\nabla^2 x_A = -2x_A$.

When we construct $\Lambda_A$ in the microscopic description, we will add terms that are total derivatives of operators made from the LLL fermion fields in order to try to remove some number of leading deformation terms $\sim g_{\nabla^{2n}\mathcal{O}} \nabla^{2n}\mathcal{O}$. Any total derivative will have no effect on the Hamiltonian, but in terms of the effect on $\widetilde{\Lambda}$ it will be equivalent to shifting the coefficient $g_{\mathcal{O}}$ of the primary operator. This means that for all practical purposes, when we construct $\Lambda_A$, without loss of generality we can treat all couplings in $\mathcal{H}$ as additional independent parameters compared to their values in the Hamiltonian![11]

---

[11] Note however that the values of the coefficients $g_{\nabla^{2n}\mathcal{O}}$ all scale like different powers of $N$, so although their contribution to $\widetilde{\Lambda}$ collapses to the combination $\left[ \sum_{n=0}^{\infty} (-2)^n g_{\nabla^{2n}\mathcal{O}} \right]$, one should keep in mind the implicit $N$-dependence of this expression when comparing different values of $N$.

To implement the procedure just outlined, we will compute the contributions to $\widetilde{\Lambda}_A$ from the terms proportional to $V_0, V_1$, and $h$ independently, and then we will fix these coefficients in order to try to remove total derivatives from $\mathcal{H} - T_0^0$. We expect that the largest corrections come from derivatives of the lowest dimension operators, so we set one condition to be that the following matrix element should vanish:

$$\langle \text{vac} | \widetilde{\Lambda}_z | \partial_A \epsilon \rangle = 0 \,, \tag{32}$$

where the state $|\partial_A \epsilon\rangle$ is the spin-1 state with dimension closest to $\Delta_\epsilon + 1 = 2.41$. For the second condition, one could impose

$$\langle \text{vac} | \widetilde{\Lambda}_z | \partial_A \partial^2 \epsilon \rangle = 0 \,, \qquad \text{or} \qquad \langle \text{vac} | \widetilde{\Lambda}_z | \partial_A \epsilon' \rangle = 0 \,, \tag{33}$$

where the state $|\partial_A \partial^2 \epsilon\rangle$ is the spin-1 state with dimension closest to $\Delta_\epsilon + 3 = 4.41$, and $|\partial_A \epsilon'\rangle$ is the spin-1 state with dimension closest to $\Delta_{\epsilon'} + 1 = 4.83$. We will choose the former condition, though in practice we have found that the difference between them is very small (about 1% in the value of $V_0$ and 0.01% in the value of $h$). Alternatively, one one could consider tuning $V_0$ and $h$ by minimizing the norm $\|\widetilde{\Lambda}_z |\text{vac}\rangle\|^2$;[12] applying this prescription would adjust the optimal value of $V_0$ by approximately 8% and $h$ by $\sim 0.03\%$, and would have only a minor impact on the results presented in section 5.1, the main difference being a slight improvement in matrix elements of $\widetilde{\Lambda}_z$ between higher-energy states and a slight increase in error for lower-energy ones. It would be interesting to explore whether different conditions can lead to parametric improvements.

Finally, using only $V_0, V_1$, and $h$, we still have one more free parameter, which is equivalent to setting the overall scale of $\widetilde{\Lambda}_z$; the reason the overall rescaling of $\widetilde{\Lambda}_A$ differs from that of the Hamiltonian $H$ is that $\mathcal{H}$ in general will contain descendants of the energy-momentum tensor, e.g. $\nabla^2 T_0^0$. We fix this overall scale by demanding

$$\langle \epsilon | \widetilde{\Lambda}_z | \partial_A \epsilon \rangle = \sqrt{2 \Delta_\epsilon} \,, \tag{34}$$

as predicted by the conformal algebra.

# 5 Numeric results

## 5.1 Matrix elements

The most direct test of our construction of the conformal generators is to compare their matrix elements between energy eigenstates with the predictions from conformal symmetry, given in (20) for scalar primaries and in appendix E for spinning primaries (see also Fig. 3). For simplicity, we will compare only the matrix elements of $\Lambda_z$ in the $j_z = 0$ sector (for scalar primaries, these are sufficient to determine all other matrix elements of $\Lambda_A$). The first comparison is shown in Fig. 7, for four scalar primaries ($\sigma, \sigma', \epsilon$ and $\epsilon'$) and their descendants. One matrix element, connecting $\epsilon$ to $\partial \epsilon$, is exact by construction since it was used to fix the linear combination of microscopic operators for $T_0^0$, but all the other matrix elements are predictions. Black dots/squares indicate respectively parity-even/-odd energy eigenstates, and are labelled by their dimension from the fuzzy sphere numerics. We depict the matrix elements of $\Lambda_z$ between these eigenstates as arrows labelled by the fuzzy sphere and CFT results for the matrix elements, as well as the relative error between the two. We show similar plots for spinning primaries in Fig. 8 and Fig. 9.

---

[12]We thank Yin-Chen He for suggesting this approach.

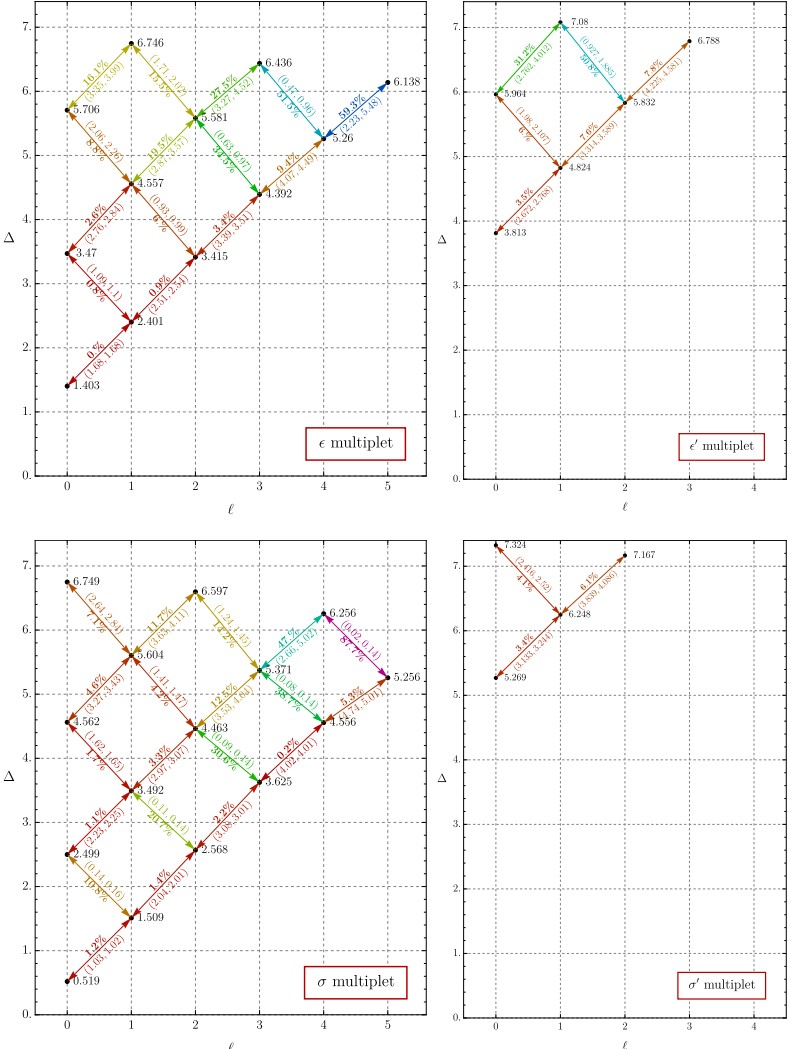

Figure 7: Quality of matrix elements of $\Lambda_z$ for various scalar primaries and their descendants. Energy eigenstates are shown as black dots (parity even) and squares (parity odd) and labelled by their dimension from the fuzzy sphere numerics. Matrix elements of $\Lambda_z$ between eigenstates are shown as arrows and labelled by the fuzzy sphere and CFT results, $(\Lambda_{z,\mathrm{FS}}, \Lambda_{z,\mathrm{CFT}})$, for the matrix elements, as well as the relative error between the two. The color of the arrows is intended to be a graphical depiction of the relative error, and varies from red to purple for small to large errors. For the CFT 'prediction', we use the formulas in appendix E and substitute the dimension of the primary from the conformal bootstrap. For descendant states, we choose the states that have the largest overlap with the state predicted by acting with $\Lambda_z$ on the eigenstates at lower levels.

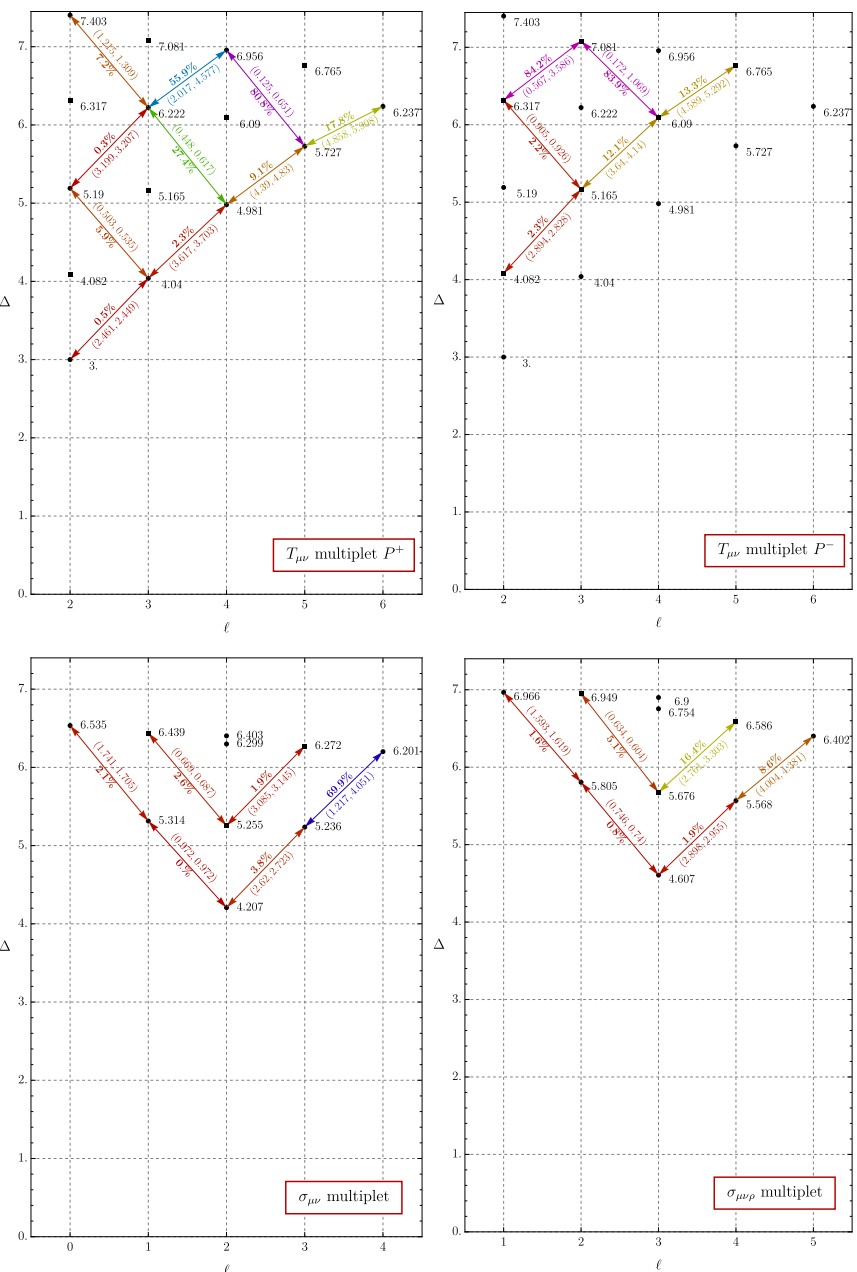

Figure 8: Quality of matrix elements of $\Lambda_z$ for various spinning primaries and their descendants. The format is the same as in Fig. 7. Parity-even *(top-left)* and parity-odd *(top-right)* descendants of $T$ are shown separately for clarity.

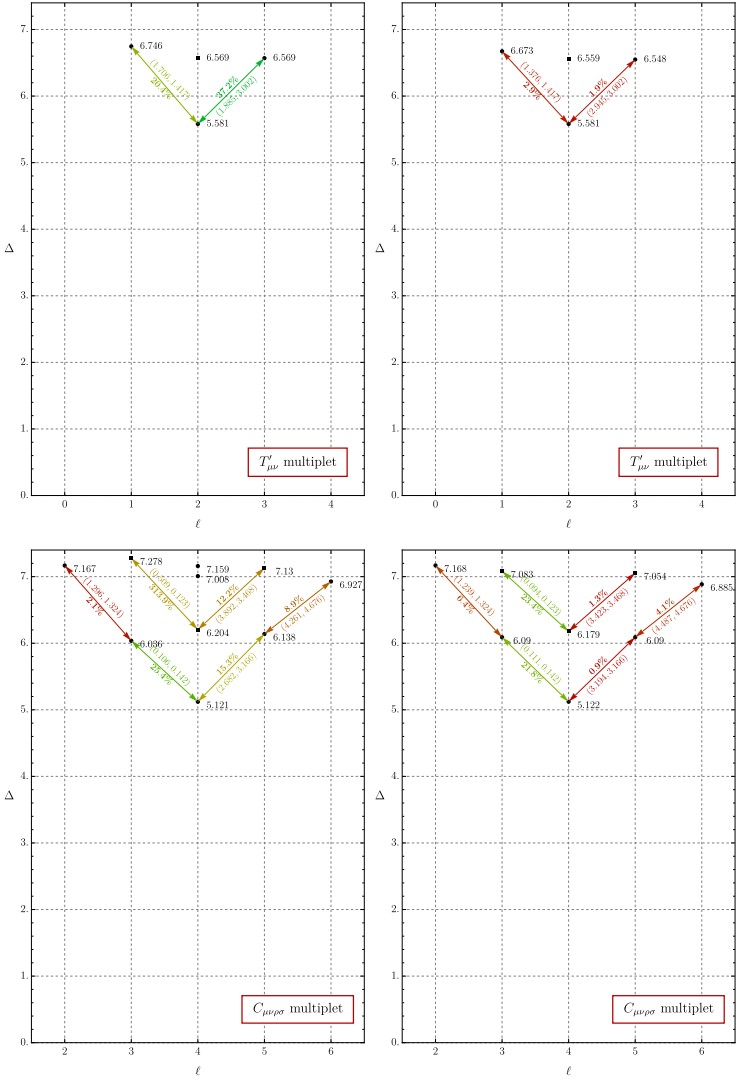

Figure 9: Quality of matrix elements of $\Lambda_z$ for the spin-2 primary operator $T'$ and some of its descendants (top), and the spin-4 primary operator $C_{\mu\nu\rho\sigma}$ and some of its descendants (bottom). The format is the same as in Fig. 7. (*Left*:) The states are all chosen to be energy eigenstates. (*Right:*) The states are chosen by first identifying the primaries $T'$ and $C$ by looking for states that are nearly annihilated by $K_A$, and then by building up the descendants by acting on these primaries with $P_A$. The primary states $T'$ and $C$ chosen in the latter case are linear combinations with large overlaps with multiple eigenstates, and so identifying the primaries by using $K$ leads to a significant improvement in the matrix elements.

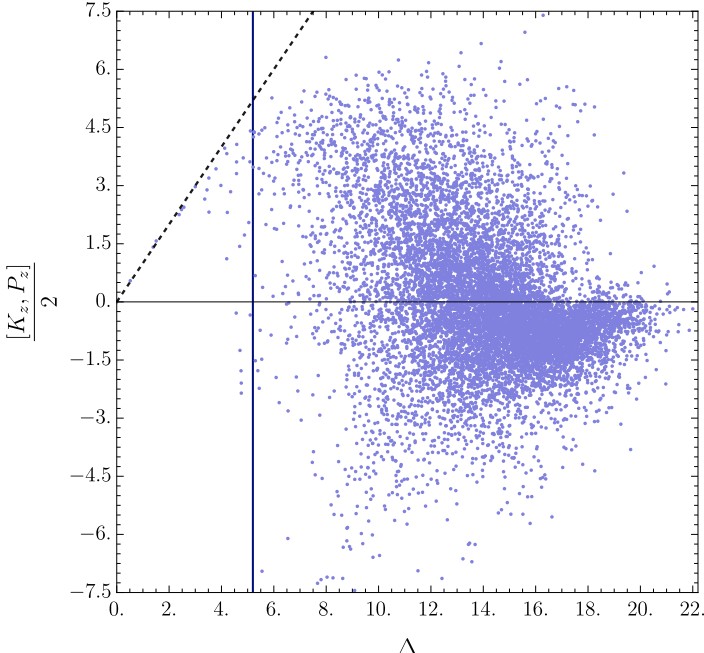

Figure 10: Quality of the commutation relation $[K_z, P_z] = 2D$ at $N = 10$, for all 5476 $j_z = 0$ states. For each eigenstate $|\Delta\rangle$, we show $\langle\Delta|[K_z, P_z]|\Delta\rangle/2$ evaluated on the state on the $y$-axis, and its dimension $\Delta$ on the $x$-axis, so that if the conformal algebra is exact then states should fall on the dashed diagonal line. A vertical line indicates roughly where the cutoff appears to be by eye.

In general, one can see that the error begins small for the low energy states, typically around 1%, and increases as one moves to descendants at higher energies. Note, however, that there are some outliers in this respect, where the errors are noticeably larger than similar matrix elements at the same dimension. In the case of the $\sigma$ multiplet, these all appear to be cases where the matrix elements themselves are unusually small, because the dimension of $\sigma$ is very close to $1/2$, the dimension of a free scalar field in $d = 3$. Consequently, $\partial^2\sigma$ is very close to being a null state, and the CFT prediction for the matrix elements is that they should be suppressed by a power of $\Delta_\sigma - 1/2$. We suspect that this suppression is what increases the sensitivity of these matrix elements to small deviations from conformality.

Another potential source of error is that conformal symmetry breaking leads to mixing among the different primaries. As a result, it is not guaranteed that descendant states will be dominantly given by any one single energy eigenstate, but instead can be a linear combination of eigenstates. In fact, we emphasize that purely looking at the energies of the states, and looking for shifts by $+1$ between descendants, does not always uniquely pick out a 'best' eigenstate at each level. Instead, the way we chose the eigenstates in these tables was by looking at the states with the largest overlap with $P_z$ acting on descendants at one level below. One of the more striking examples of this effect is shown in Fig. 9, where we consider the $T'$ (spin-2, dimension $\approx 5.6$) and $C$ (spin-4, dimension $\approx 5.0$) primaries. In both of these cases, we find that even the matrix elements of $\Lambda_z$ between the primary and its level-1 descendants has rather large errors. As we will discuss in section 5.3, the main source of error here appears to be the fact that at $N = 16$, the primary state itself is not dominantly an energy eigenstate, but rather is much more accurately described as a linear combination of multiple energy eigenstates.

## 5.2 Commutator

Another fairly direct test of our construction of the conformal generators is how accurately they satisfy the conformal algebra. Such a test is similar to the direct test of the matrix elements in the previous subsection, but has a conceptual advantage that it is relies less heavily on the specific choice of basis. In other words, one can consider an operator equation such as

$$[K_z, P_z] = 2D, \tag{35}$$

and evaluate it between any bra and ket states one chooses.

Because acting with $P_z$ raises the energy of states, to obtain an accurate calculation of the commutator (35) in a given state $|\psi\rangle$, one needs to obtain eigenstates $|n\rangle$ with energies $\sim +1$ larger than that of $|\psi\rangle$:

$$\langle\psi|[K_z, P_z]|\psi\rangle = \sum_n |\langle n|P_z|\psi\rangle|^2 - \sum_n |\langle n|K_z|\psi\rangle|^2. \tag{36}$$

When the energy of $|\psi\rangle$ is small, obtaining such eigenstates is not an issue. But when the energy of $|\psi\rangle$ is large $\sim O(7)$, numerically finding all eigenstates with energies up to $\Delta_\psi + 1$ can be significantly more computationally expensive than finding all eigenstates with energies up to $\Delta_\psi$. Therefore, our numeric results for the commutator are affected by an unphysical 'eigenvalue cutoff' which arises purely from the fact that we only find a subset of all eigenvalues of the (finite dimensional) fuzzy sphere Hilbert space. At $N = 10$, the total dimension of the fuzzy sphere Hilbert space in the $j_z = 0$ sector is dim = 5476, which is small enough to fully diagonalize the Hamiltonian and so avoid having to introduce an eigenvalue cutoff. In this case, we show in Fig. 10 the expectation value of the commutator $[K_z, P_z]/2$ for all $\mathbb{Z}_2$-even eigenstates, versus their dimension $\Delta$. At low dimensions, one sees that $[K_z, P_z]/2$ is very close to $\Delta$, as predicted by the conformal algebra. However, as the dimension grows, the errors become large, and most of the values of $[K_z, P_z]$ are far from the CFT prediction starting at $\Delta \sim 5.25$, as indicated by a vertical line in the figure. We take this to be an optimistic estimate of the UV cutoff of the CFT description, above which the fuzzy sphere regulator becomes significant and leads to large corrections to the CFT description.

Another qualitative feature of Fig. 10 that can be readily understood is that once the deviations from the CFT algebra become large, there are many negative values of $[K_z, P_z]$. Because the size of the Hilbert space is finite, the trace of the commutator must vanish, which means that the sum over the $y$-values of all the dots in the figure must identically add up to zero. Since by unitarity, $[K_z, P_z]$ must be positive in the regime where the CFT algebra holds, this low energy regime creates a deficit of positive values that must be compensated for at high energies by negative values.

More generally, we can look at the quality of the commutator at various values of $N$. In Fig. 11, we show the same commutator plot for $N = 6, 8, 10, 12, 14$ and 16. Up to $N \leq 10$, we obtain all eigenstates of the fuzzy sphere Hamiltonian, so there is no effect from an eigenvalue cutoff (though $x$-axis of the plots themselves only go up to $\Delta = 7$, all intermediate states are included when evaluating the commutator). For $N = 12, 14$ and 16, we obtain all eigenstates up to $\Delta = 9.06, 9.03, 7.44$, respectively. We also show a vertical line on each plot indicating a rough estimate of the cutoff. To obtain this estimate, we take our estimate $\Lambda_{UV} \approx 5.25$ from $N = 10$ and in each plot we rescale it proportionally to $\sqrt{N}$:

$$\Lambda_{UV} \approx 1.66\sqrt{N}. \tag{37}$$

The cutoff obtained this way lines up by eye with where one sees the corrections to the commutators become large at the various values of $N$.

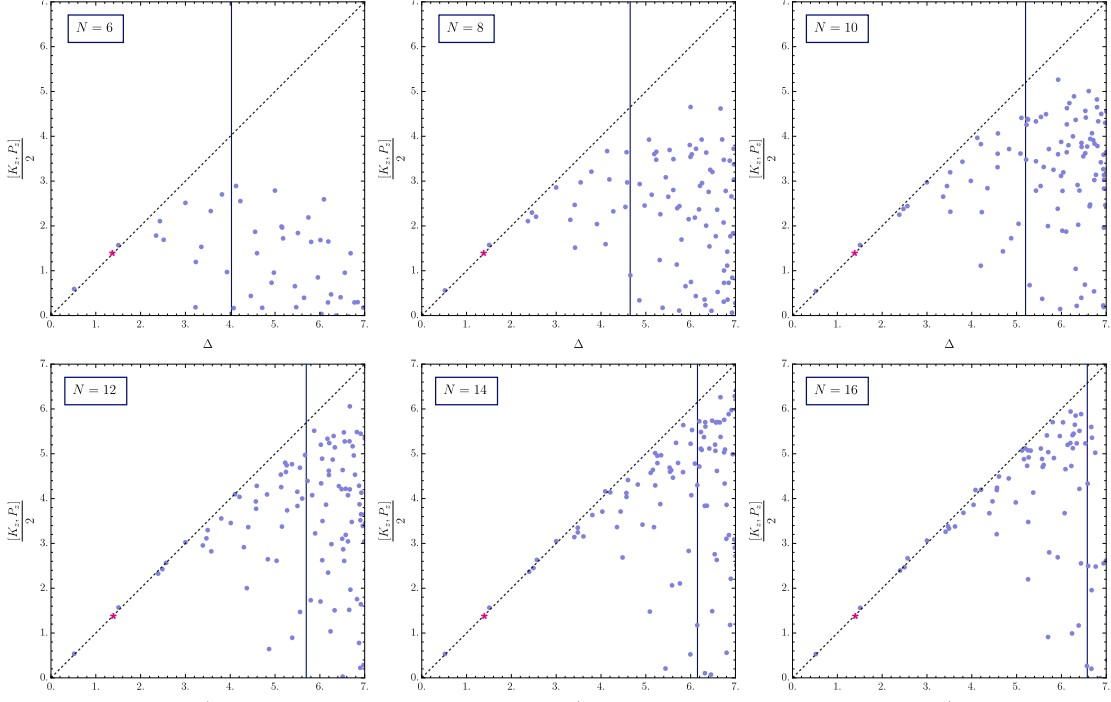

Figure 11: Quality of the commutation relation $[K_z, P_z] = 2D$, as in Fig. 10, for $N = 6, 8, 10, 12, 14, 16$. In computing the commutator, we keep all states on the fuzzy sphere for $N = 6, 8, 10$, but only up to $\Delta = 9.06, 9.03, 7.44$ respectively for $N = 12, 14, 16$. The magenta asterisk represents the point fixed by the condition (34). The vertical line in each plot is the cutoff chosen at $N = 10$ from Fig. 10, but scaled by $\sqrt{N}$; note that this scaling lines up reasonably well with where the accuracy of the commutator starts to fail by eye at different values of $N$.

### 5.3 Identifying primaries

One of the motivations for constructing the conformal generators is that we would like to use them to identify primary states with a more precise method than looking at integer shifts in the dimensions of operators, since the latter method becomes intractable when the spectrum is very dense. Moreover, when the spectrum is dense, even small deviations from the critical point will lead to large operator mixing, and one might hope that states can still be divided into primaries and descendants if one looks at general linear combinations of eigenstates. In this subsection, we will see that indeed this is possible. In particular, we will try to identify primary operators based on the condition that they are approximately annihilated by $K_\pm$ and $K_z$, or equivalently that

$$\langle \psi ||K^2|| \psi \rangle \equiv \sum_{A=1}^{3} \langle \psi | K_A^\dagger K_A | \psi \rangle \ll 1 \,. \tag{38}$$

Crucially, $|\psi\rangle$ here does not need to be an energy eigenstate. Rather, the way we will find primaries is by diagonalizing the matrix $|K^2|$ and looking for small eigenvalues.

An important consideration when interpreting these eigenstates of $|K^2|$ is that we have already seen the matrix elements of $K$ start to deviate significantly from the CFT predictions at high energies. Consequently, for each potential primary state that we find among the small eigenvalues of $|K^2|$, we also test whether or not the conformal commutator $[K_z, P_z] = 2D$ is still accurately reproduced on that state. We present these results in Fig. 12 at $N = 16$, where

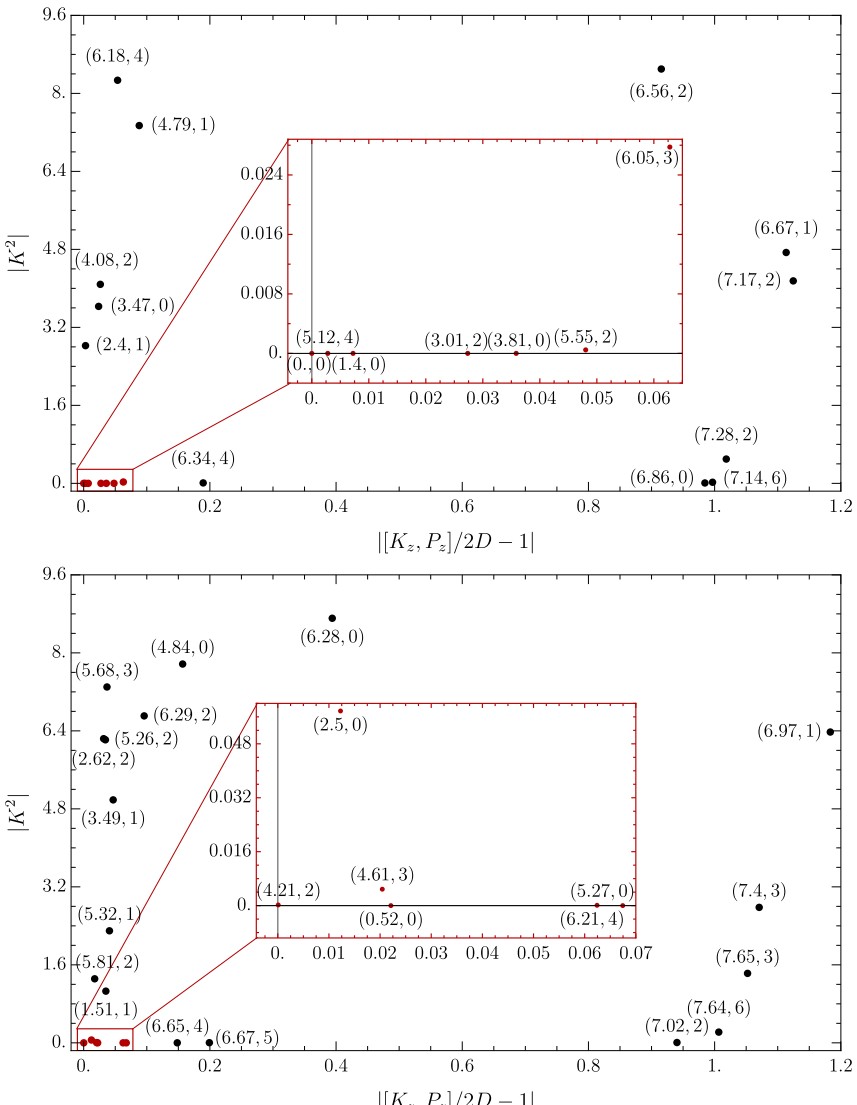

Figure 12: Eigenstates of $|K^2|$, for $\mathbb{Z}_2$-even (*top*) and $\mathbb{Z}_2$-odd (*bottom*). Each dot represents an eigenstate labelled by its $(\Delta, \ell)$, and its location indicates its $|K^2|$ eigenvalue as well as the error in the $[K_z, P_z]$ commutator evaluated on the state. Primaries should have $|K^2| \ll 1$, and moreover we demand that there is a small error in the commutator in order to test that the value of $|K^2|$ is reliable. The lower-left region which contains such primaries is shown magnified. As discussed in the body of the text, the $(\Delta, \ell) = (6.05, 3)$ 'almost primary state' in the upper plot is a spin-3 descendant $\sim \partial_\mu C^{\mu\nu\rho\sigma}$ of $C$ that is almost null because $C$ is almost a conserved higher-spin current; similarly, the $(\Delta, \ell) = (2.5, 0)$ state in the bottom plot is $\partial^2 \sigma$, which is almost null because $\Delta_\sigma \approx 1/2$. See Appendix G for more precise values of conformal dimensions.

all eigenstates of $|K^2|$ are shown as dots labeled by their dimension[13] and spin $(\Delta, \ell)$ and located according to their eigenvalue under $|K^2|$ and their error in the $[K_z, P_z]$ commutator; reliable primaries should have both small values for $|K^2|$ and for the error in the commutator, and therefore appear in the lower left-hand corner. We have magnified this lower left-hand corner, so that one can see there are a subset of states identified as primaries by this method.

Interestingly, this approach is still too naive in one important respect, which one can see by inspection of the figure. The problem is that the magnified region, both in the $\mathbb{Z}_2$-even and $\mathbb{Z}_2$-odd sectors, contains one interloper that is actually a descendant. In the $\mathbb{Z}_2$-even sector, this interloper is the $(\Delta, \ell) = (6.05, 3)$ descendant of $C^{\mu\nu\rho\sigma}$, and in the $\mathbb{Z}_2$-odd sector, it is the $(\Delta, \ell) = (2.5, 0)$ descendant of $\sigma$. In both cases, note that the interloper still has a significantly larger value of $|K|^2$ than the true primaries, and so could potentially be identified as a descendant just by comparison with the other states. However, there is a much more robust signal that these states are actually descendants. The key point is that the reason these states have small values of $|K|^2$ is not numeric or truncation errors, but rather because $|K|^2$ for these states actually is small in the CFT! In the case of the spin-3 descendant of $C^{\mu\nu\rho\sigma}$, the reason is that $C^{\mu\nu\rho\sigma}$ has dimension $\Delta = 5.02$ and therefore is very nearly a higher-spin conserved current, which would imply that its level-1 spin-3 descendant $\partial_\mu C^{\mu\nu\rho\sigma}$ is null. Referring to Fig. 3, one can check that with $\Delta_0 = 5.0226$ and $\ell_0 = 4$, the size of $|K_z|^2$ should be $\frac{2\ell_0(\Delta_0 - \ell_0 - 1)}{2\ell_0 + 1} = 0.02$, and therefore $|K^2|$ should be $\frac{2\ell_0 + 1}{\ell_0}|K_z|^2 = 0.045$, roughly the size of $|K^2|$ from the figure. Similarly, for the level-2 spin-0 descendant $\partial^2 \sigma$, the size of $|K^2|$ should be $3|K_z|^2 = 2(\Delta_0 - 1) = 0.072$ for $\Delta_0 = 0.518$, again close to the value in the figure. Therefore, since the action of $K_A$ on these states tells us which states they would be descendants of if they were descendants, it is it straightforward to see that $|K^2|$ is not small when compared to the CFT prediction.

Finally, we can now return to an issue we noted in section (5.1), namely that the quality of the matrix elements of $\Lambda_z$ in the plots on the left of Fig. 9 are worse than is typical for states at the corresponding dimension. The resolution is that when we look at small eigenvalues of $|K^2|$ to identify the primary states $C$ and $T'$, we find that they are not dominantly made out of any single energy eigenstate, but rather are linear combinations of multiple eigenstates. When we recompute the matrix elements using primaries identified with $|K^2|$, and descendants constructed by acting on them with $P_A$, we obtain the plots on the right, where the quality of the matrix elements is noticeably improved.

## 5.4 Conformal Casimir

In principle, a simple way to immediately group all states into conformal representations is to compute the quadratic casimir $\mathcal{C}_2$ for the conformal algebra,

$$\mathcal{C}_2 = D^2 + L^2 - \frac{1}{2}\{K_A, P_A\} = D^2 + \ell(\ell + 1) - \frac{1}{2}\{K_A, P_A\} \tag{39}$$

$$= D(D - 3) + \ell(\ell + 1) - P_A K_A.$$

In the first line, we set $L^2 \to \ell(\ell + 1)$ because we work with eigenstates of $L^2$. By contrast, $P_A$ and $K_A$ change the state, and the expression we get in the second line uses their commutator and so is equivalent to the first line only in the CFT limit. Nevertheless, this last expression is more convenient to use, because for a given eigenstate, computing $P_A$ on that state requires computing eigenstates with higher dimension whereas computing $K_A$ only requires computing eigenstates with lower dimensions, and obtaining more eigenstates with higher energies is computationally expensive.

---

[13] For a general state $|\psi\rangle$, we define its dimension to be $\Delta \equiv \langle\psi|D|\psi\rangle$.

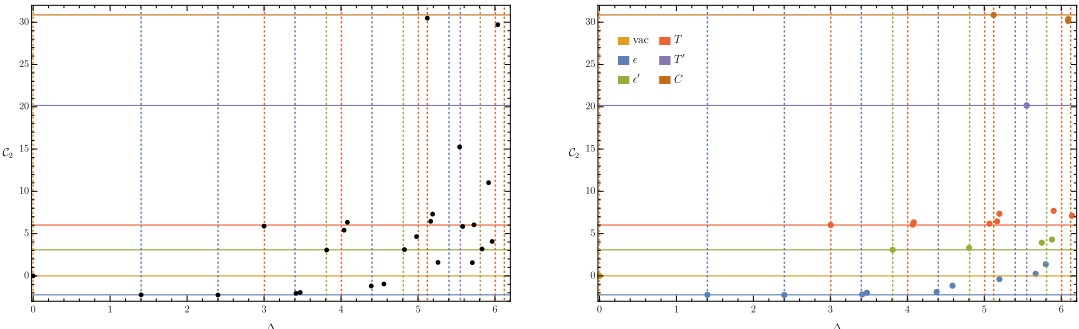

Figure 13: Conformal Casimir $\mathcal{C}_2$ and dimension $\Delta$ of $\mathbb{Z}_2$-even states, up to $\Delta \lesssim 6$, evaluated using (39). Solid horizontal lines indicate the Casimir $\mathcal{C}_2$ of a primary, and dashed vertical lines indicate the dimensions $\Delta + \mathbb{N}$ of the primary and its descendants. *(Left:)* Black dots indicate individual energy eigenstates. *(Right:)* Dots indicate states built up using $P_A$, starting with primaries identified by acting with $P_A K_A$, and then subsequently building up the descendants by acting with $P_A$. In this case, the color of the dot indicates the primary it was built from.

In Fig. 13, we show the Casimir for all $\mathbb{Z}_2$ even eigenstates with dimensions up to $\Delta < 6$, versus their dimension. For comparison with the expected results in the conformal limit, for each $\mathbb{Z}_2$-even primary state (as identified in the previous subsection by looking for states annihilated by $K_A$) we also show a solid horizontal line at the corresponding value of the Casimir $\mathcal{C}_2$ as well as a series of vertical lines at the corresponding dimensions of the primary and its descendants. Note that at low dimensions, where the matrix elements of $P_A$ are still accurate and the primary states are still well-aligned with energy eigenstates, we see a collapse of many states in the same conformal multiplet onto the same value of the conformal Casimir.

As one goes higher up in the spectrum, energy eigenstates tend to be linear combinations from multiple different conformal representations which causes their Casimir value to lie between the Casimirs of the primaries. For instance, in the exact conformal theory, the $T'$ operator has spin 2 and dimension 5.51, so $\mathcal{C}_2 = 19.8$, yet in Fig. 13 there are no states on the corresponding horizontal line. Instead, there are two nearby eigenstates with $\Delta = 5.54$ and 5.58 which are strongly mixed between $T'$ and $\partial\partial\epsilon$ (spin 2). One can mitigate this effect by working with states selected by using $P_A$ and $K_A$ rather than with energy eigenstates. Specifically, we can start with linear combinations identified as primaries in the previous section as states that are (nearly) annihilated by $K_A$. Then, we can successively build up all the other states in the conformal representation by raising with $P_A$. In Fig. 13, we also show the Casimir evaluated on states, but now for states built up this way. Because every state is constructed by repeated actions of $P_A$ on a specific primary, we also color-code the states to indicate which primary they were built from. Mostly, the difference between the two plots is minor, but in a few cases such as the $T'$ state there is a significant improvement in the value of the Casimir; indeed, it is remarkable that despite the large mixing effects on the $T'$ operator as an energy eigenstate, there still exists to very high accuracy a well-defined primary state as defined by the action of $K_A$.

# Acknowledgments

We are grateful to Yin-Chen He, Wei Li, and Slava Rychkov for discussions, and to Yin-Chen He for comments on a draft. Numeric results in this paper for $N \geq 12$ were obtained by implementing our construction within the publicly available Julia code provided at https://www.fuzzified.world.

**Funding information**   GF, ALF, and EK are supported by the US Department of Energy Office of Science under Award Number DE-SC0015845, and GF was partially supported by the Simons Collaboration on the Non-perturbative Bootstrap.

# A   Conformal generators on $S^2$ from $T^{\mu\nu}$

Here we briefly review how to derive the conformal generators on $\mathbb{R} \times S^2$ as integrals of the CFT energy-momentum tensor $T^{\mu\nu}$. We begin with the theory on the Euclidean plane $\mathbb{R}^3$, where all conformal transformation can be written in terms of a vector $\epsilon^i(x)$ ($i = 1, 2, 3$) that is a quadratic polynomial in $x^i$ (where $x \in \mathbb{R}^3$), and then transform to radial coordinates. In Euclidean signature, the time coordinate is set by the magnitude of $x^i$:

$$x^i = e^{t_E}\hat{x}^i, \qquad \sum_{i=1}^3 (\hat{x}^i)^2 = 1. \tag{A.1}$$

Denote the $\mathbb{R} \times S^2$ coordinates by $\{\xi^\mu\}_{\mu=0,1,2}$, where $\xi^0 = t_E$ is time and $\{\xi^a\}_{a=1,2}$ parameterize the unit sphere $S^2$. Denote the Jacobian for the transformation to $\mathbb{R} \times S^2$ by

$$\frac{dx^i}{d\xi^\mu}, \qquad V_\mu \equiv \frac{dx^i}{d\xi^\mu}V^i. \tag{A.2}$$

In general, the generator of a transformation parameterized by $\epsilon^i(x)$ is given by the following integral of the energy-momentum tensor:

$$Q_\epsilon = \int d^2\xi \sqrt{g}\,\epsilon_i(x)T^{0i} = \int d^2\xi \sqrt{g}\,\epsilon_i(\xi)\frac{dx^i}{d\xi^\mu}T^{0\mu}(\xi), \tag{A.3}$$

where $d^2\xi = d\xi_1 d\xi_2$ and $g_{ab}$ is the metric on $S^2$.

For dilatations,

$$\text{Dilatations: } \epsilon^i(x) = x^i \Rightarrow \epsilon_i(x)\frac{dx^i}{d\xi^\mu}T^{0\mu} = \frac{1}{2}\frac{de^{2t_E}}{d\xi^\mu}T^{0\mu} = e^{2t_E}T^{00} = T^0_{\ 0}, \tag{A.4}$$

and so the Dilatation operator is just the Hamiltonian on the sphere.

For an $SO(3)$ rotation $J_i$ around the vector $\omega^i$, $\epsilon_i(x) = \epsilon_{ijk}\omega^j x^k$:

$$\text{Rotations: } \epsilon_i(x) = \epsilon_{ijk}\omega^j x^k \Rightarrow \epsilon_i(x)\frac{dx^i}{d\xi^\mu}T^{0\mu} = 2\epsilon_{ijk}\omega^j\hat{x}^k\frac{\partial \hat{x}^i}{\partial \xi^a}g^{ab}T^0_{\ b}, \tag{A.5}$$

where the sum on $a$ is over $a = 1, 2$, and the dependence on $T^{00}$ drops out.

For a translation in the $a^i$ direction, $\epsilon^i(x) = a^i$:

$$\text{Translations: } \epsilon^i(x) = a^i \Rightarrow \epsilon_i(x)\frac{dx^i}{d\xi^\mu}T^{0\mu} = e^{-t_E}\left(a \cdot \hat{x}T^0_{\ 0} + a_i\frac{\partial \hat{x}^i}{\partial \xi^a}g^{ab}T^0_{\ b}\right). \tag{A.6}$$

Finally, special conformal transformations (SCTs) can be parameterized by a vector $b^i$, with $\epsilon^i(x) = 2(x \cdot b)x^i - b^i x^2$:

$$\text{SCTs: } \epsilon^i(x) = 2(x \cdot b)x^i - b^i x^2 \Rightarrow \epsilon_i(x)\frac{dx^i}{d\xi^\mu}T^{0\mu} = e^{t_E}\left(b \cdot \hat{x}T^0_{\ 0} - b_i\frac{\partial \hat{x}^i}{\partial \xi^a}g^{ab}T^0_{\ b}\right), \tag{A.7}$$

where we have used $\partial x^i/\partial \xi^0 = x^i$ and $x^i \partial x^i/\partial \xi^a = 0$.

The structure of these expressions are perhaps more transparent if we write them in terms of embedding space coordinates $\{\widehat{x}^A\}_{A=1,2,3}$ for the sphere, $\sum_A(\widehat{x}^A)^2 = 1$. Then, the Jacobian $\frac{\partial \widehat{x}^A}{\partial \xi^a}$ is just the factor for the uplift of a vector from $S^2$ to $\mathbb{R}^3$:

$$P_a^A \equiv \frac{d\widehat{x}^A}{d\xi^a}, \qquad T^{0A} \equiv P_a^A g^{ab} T^0_{\ b}. \tag{A.8}$$

One can define a projector $U^{AB}$ that projects tensors onto the tangent space of $S^2$:

$$U^{AB} \equiv \delta^{AB} - \widehat{x}^A \widehat{x}^B. \tag{A.9}$$

With this notation, the generators are integrals of the following forms:

$$Q_\epsilon = \int d^2\Omega \mathcal{I}_\epsilon,$$

$$\text{Dilatations: } \mathcal{I}_\epsilon = T^0_{\ 0},$$

$$\text{Rotations } \omega^A : \mathcal{I}_\epsilon = 2\epsilon_{ABC}\omega^B \widehat{x}^C T^{0A}, \tag{A.10}$$

$$\text{Translation } a_A : \mathcal{I}_\epsilon = e^{-t_E} a_A (\widehat{x}^A T^0_{\ 0} + T^{0A}),$$

$$\text{SCTs } b_A : \mathcal{I}_\epsilon = e^{t_E} b_A (\widehat{x}^A T^0_{\ 0} - T^{0A}).$$

Ultimately, we want to use formulas for the generators in Lorentzian signature rather than Euclidean signature, so we have to Wick rotate $t_E = it_L$ under which $T^0_{\ 0} \to T^0_{\ 0}$ and $T^0_{\ A} \to iT^0_{\ A}$.

Finally, we will generally take our intrinsic coordinates on the sphere to be spherical coordinates $\theta, \phi$, with $d^2\Omega \equiv d\phi\, d\cos\theta$, and we will group the generators into combinations with definite values of $J_3$. For rotations, these are the standard combinations $J_z = J_3$ and $J_\pm = J_1 \pm iJ_2$:

$$J_z = \int d^2\Omega T^0_{\ \phi}, \qquad J_\pm = \pm i \int d^2\Omega e^{\pm i\phi}(T^0_{\ \theta} \pm i\cot\theta\, T^0_{\ \phi}). \tag{A.11}$$

For translations and SCTs, we can take $P_z, P_\pm = P_1 \pm iP_2$ and $K_z, K_\pm = K_1 \pm iK_2$. Setting $t_L = 0$ for simplicity,

$$P_z = \int d^2\Omega(\cos\theta\, T^0_{\ 0} - i\sin\theta\, T^0_{\ \theta}), \quad P_\pm = \int d^2\Omega e^{\pm i\phi}\sin\theta(T^0_{\ 0} + i(\cot\theta\, T^0_{\ \theta} \pm i\csc^2\theta\, T^0_{\ \phi})),$$

$$K_z = \int d^2\Omega(\cos\theta\, T^0_{\ 0} + i\sin\theta\, T^0_{\ \theta}), \quad K_\pm = \int d^2\Omega e^{\pm i\phi}\sin\theta(T^0_{\ 0} - i(\cot\theta\, T^0_{\ \theta} \pm i\csc^2\theta\, T^0_{\ \phi})). \tag{A.12}$$

# B  Rotation Noether current $V_a$

The microscopic fermionic theory has time-translation invariance and rotational invariance, with the following corresponding Noether charge densities, respectively:

$$\mathcal{H} = \left(\sum_n g_n(\psi^\dagger\psi)\nabla^{2n}(\psi^\dagger\psi) - g_{n,z}(\psi^\dagger\sigma^z\psi)\nabla^{2n}(\psi^\dagger\sigma^z\psi)\right) - h\psi^\dagger\sigma^x\psi, \tag{B.1}$$

$$V_a \propto \psi^\dagger iD_a\psi.$$

In this work, we do not use $V_A$ as part of the construction of the conformal generators, and in this appendix we will make some comments about why. To an extent, the reason is simply that it was not necessary to use it. However, there is also an interesting effect that its contribution to $P_A$ and $K_A$, through $\int d^2\Omega V_A$, turns out to vanish. Consequently it is not useful to construct the CFT stress tensor $T^0_A$ components by starting with $V_A$ and adding small corrections.

To see this, first expand out the definition of $V_A$:

$$V_A \equiv P^A_a g^{ab} V_b \propto \frac{\partial \widehat{x}^A}{\partial \xi^a} g^{ab} \left[ \psi^\dagger (\overleftrightarrow{\partial}_b + 2isA_b)\psi \right]. \tag{B.2}$$

When the fermions are restricted to the LLL, $V_A$ simplifies due to the following identity:

$$\frac{\partial \widehat{x}^A}{\partial \xi^a} g^{ab} i \left[ \Phi^*_m(\overleftarrow{\partial}_b - \overrightarrow{\partial}_b + 2isA_b)\Phi_{m'} \right] = i\zeta^{(A)a}\partial_a(\Phi^*_m\Phi_{m'}), \tag{B.3}$$

where $\zeta^{(A)}$ are the Killing vectors for the rotations on the sphere:

$$\zeta^{(1)a}\partial_a = \sin\phi\,\partial_\theta + \cot\theta\cos\phi\,\partial_\phi\,,$$

$$\zeta^{(2)a}\partial_a = -\cos\phi\,\partial_\theta + \cot\theta\sin\phi\,\partial_\phi\,, \tag{B.4}$$

$$\zeta^{(3)a}\partial_a = -\partial_\phi\,.$$

Consequently, restricted to the LLL states

$$V^A \propto \zeta^{(A)a}\partial_a n^0\,, \qquad n^0 \equiv \psi^\dagger\psi\,. \tag{B.5}$$

When we consider the generators of rotations, we manifestly get quantities that simply compute the sum of the corresponding quantities for the individual fermions, as we would expect since rotations are exactly preserved on the fuzzy sphere:

$$J^A = \frac{s+1}{2}\epsilon_{ABC}\int d^2\Omega \widehat{x}^C\zeta^{(B)a}\nabla_a n^0 = -\frac{s+1}{2}\epsilon_{ABC}\int d^2\Omega n^0\zeta^{(B)a}\nabla_a\widehat{x}^C$$

$$= (s+1)\int d^2\Omega \widehat{x}^A n^0\,, \tag{B.6}$$

where we have used $\zeta^{(B)a}\nabla_a = \epsilon^{BDE}\widehat{x}^D\partial_E$ and $\epsilon_{ABC}\epsilon^{BDC} = -2\delta^D_A$.

Now we come to the generators $P_A - K_A$. In this case one finds that if we try to build the generator by taking $T^{0A} \sim V^A$, then the integrand is a total derivative and therefore the result vanishes:

$$(P_A - K_A) \overset{?}{\approx} \int d^2\Omega V_A \propto \int d^2\Omega \zeta^{(B)a}\nabla_a n^0 = 0\,. \tag{B.7}$$

One puzzling aspect of this relation is that it seems to violate the standard commutation relations for the rotation Noether current. More precisely, the anticommutation relations

$$\{\psi^\dagger(t,\widehat{x}), \psi(t,\widehat{y})\} = i\delta^{(2)}(\widehat{x},\widehat{y})\,, \tag{B.8}$$

where $\delta^{(2)}(x,y)$ is the $\delta$ function on the sphere, imply that $V_a$ has the following commutator,

$$\left[ V_a(x), (\psi^\dagger\psi)(y) \right] \propto -2(\psi^\dagger\psi)(x)\partial_a\delta(x,y)\,. \tag{B.9}$$

Inserting a factor of $\widehat{x}$ and integrating, this should imply that $P_A - K_A$ built from $V_a$ cannot vanish.

In fact, this issue was noted as early as [23] in the context of the LLL on the plane, and it was traced back to the modification of the fermion anticommutation relations due to the projection onto the LLL. After this projection, the fermion anticommutation relations are nonlocal:

$$\{\psi^\dagger(\widehat{x}), \psi(\widehat{y})\} \rightarrow \sum_{m=-s}^{s} \Phi_m^*(\widehat{x})\Phi_m(\widehat{y}) \neq i\delta_{S^2}(x-y). \tag{B.10}$$

In [23], a prescription was given for constructing a modified $V_a$, essentially by integrating out the higher Landau levels using the equations of motion, in order to restore the conservation equation for translations in flat space. In practice, this procedure essentially constructs the $V_a$ components of the CFT stress tensor as an infinite sum over local operators designed to reproduce the correct Ward identities for the IR CFT, similarly to how we build $T_0^0$.

## C  Formulae for $H$ and $\Lambda_A$

In this section we briefly summarize how to write the Hamiltonian in second quantized form as done in [6] and how to extend this formalism to the other conformal generators. Recall that fermions defined on the LLL can be expanded as the product of creation/annihilation operators and monopole harmonics

$$\psi(\theta,\phi) = \sum_{m=-s}^{s} \Phi_m(\theta,\phi)\mathbf{c}_m, \qquad \psi^\dagger(\theta,\phi) = \sum_{m=-s}^{s} \Phi_m^*(\theta,\phi)\mathbf{c}_m^\dagger, \tag{C.1}$$

with $\mathbf{c}_m^\dagger \equiv (c_{m\uparrow}^\dagger, c_{m\downarrow}^\dagger)$. To express this observable in terms of $\mathbf{c}$s, it is helpful to recall certain properties of monopole and spherical harmonics. Both functions are special cases of the more general spin-weighted spherical harmonics $Y_{\ell,m}^{(s)}$ ($\ell = s, s+1, \cdots$)

$$Y_{\ell,m}^{(s)} = (-1)^{\ell+m-s}\sqrt{\frac{(\ell+m)!(\ell-m)!(2\ell+1)}{4\pi(\ell+s)!(\ell-s)!}}e^{im\phi}\sin^{2\ell}\left(\frac{\theta}{2}\right)$$
$$\times \sum_{r=0}^{\ell-s}(-1)^r\binom{\ell-s}{r}\binom{\ell+s}{r+s-m}\cot^{2r+s-m}\left(\frac{\theta}{2}\right), \tag{C.2}$$

from which the usual spherical harmonic $Y_{\ell,m}$ and the monopole harmonics $\Phi_m$ are recovered as

$$Y_{\ell,m} = Y_{\ell,m}^{(0)}, \qquad\qquad Y_{\ell,m}^* = (-1)^m Y_{\ell,-m}^{(0)},$$
$$\Phi_m = e^{i\pi(s-2m)}Y_{s,m}^{(-s)}, \qquad \Phi_m^* = e^{i\pi m}Y_{s,-m}^{(s)}. \tag{C.3}$$

Spin-weighted harmonics satisfy the following useful properties

$$\int_{S^2} Y_{\ell,m}^{(s)}Y_{\ell',m'}^{(s)*} = \delta_{\ell,\ell'}\delta_{m,m'}, \tag{C.4a}$$

$$\sum_{\ell,m} Y_{\ell,m}^{(s)}(\Omega_1)Y_{\ell,m}^{(s)*}(\Omega_2) = \delta^{(2)}(\Omega_1-\Omega_2), \tag{C.4b}$$

$$\int_{S^2} Y_{\ell_1,m_1}^{(s_1)}Y_{\ell_2,m_2}^{(s_2)}Y_{\ell_3,m_3}^{(s_3)} = \sqrt{\frac{\prod_{i=1}^3(2\ell_i+1)}{4\pi}}\begin{pmatrix}\ell_1 & \ell_2 & \ell_3\\ m_1 & m_2 & m_3\end{pmatrix}\begin{pmatrix}\ell_1 & \ell_2 & \ell_3\\ -s_1 & -s_2 & -s_3\end{pmatrix}. \tag{C.4c}$$

Let us define

$$n_i(\Omega) \equiv \psi^\dagger(\Omega) n_i \psi(\Omega), \tag{C.5}$$

with $n_0 = 1$ and $n_{x,y,z} = \sigma_{x,y,z}$ the usual Pauli matrices, then we can write the Hamiltonian

$$H = \int d\Omega_1 \mathcal{H}, \tag{C.6}$$

$$\mathcal{H} = \left[ \int d\Omega_2 U(\Omega_{12}) (n_0(\Omega_1) n_0(\Omega_2) - n_z(\Omega_1) n_z(\Omega_2)) \right] - h n_x(\Omega_1), \tag{C.7}$$

$$U(\Omega_{12}) = \frac{\lambda_0}{R^2} \delta(\Omega_1 - \Omega_2) + \frac{\lambda_1}{R^4} \nabla^2 \delta(\Omega_1 - \Omega_2). \tag{C.8}$$

Now we can plug in the decomposition (C.1), then the part proportional to $h$ is straightforward to compute using the orthonormality of monopole harmonics

$$\int \psi^\dagger \sigma_x \psi = \sum_{m_1,m_2} \mathbf{c}^\dagger_{m_1} \sigma_x \mathbf{c}_{m_2} \underbrace{\int \Phi^*_{m_1} \Phi_{m_2}}_{\delta_{m_1,m_2}} = \sum_m \mathbf{c}^\dagger_m \sigma_x \mathbf{c}_m. \tag{C.9}$$

For the remaining parts, let us define $\sigma_\alpha$, $\alpha = 0, z$, and let us rewrite

$$\sum_{m_i} (\mathbf{c}^\dagger_{m_1} \sigma_\alpha \mathbf{c}_{m_4})(\mathbf{c}^\dagger_{m_2} \sigma_\alpha \mathbf{c}_{m_3}) \int d\Omega_1 d\Omega_2 U(\Omega_{12}) \Phi^*_{m_1}(\Omega_1) \Phi_{m_4}(\Omega_1) \Phi^*_{m_2}(\Omega_2) \Phi_{m_3}(\Omega_2), \tag{C.10}$$

$$U(\Omega_{12}) = \sum_{\ell=0}^\infty \sum_{m=-\ell}^\ell \beta_i(\ell) Y_{\ell,m}(\Omega_1) Y^*_{\ell,m}(\Omega_2), \qquad \beta_i(\ell) = \begin{cases} \beta_0 = \frac{\lambda_0}{R^2}, \\ \beta_1 = -\ell(\ell+1)\frac{\lambda_1}{R^4}, \end{cases} \tag{C.11}$$

where we have rewritten $\delta(\Omega_1 - \Omega_2)$ as in (C.4b) and used the fact that $\nabla^2 Y_{\ell,m} = -\ell(\ell+1) Y_{\ell,m}$. Focusing on the integral, we can use the definition in terms of spin-weighted harmonics in (C.3) together with the expression in (C.4c)

$$\mathcal{I}^{(i)} \equiv \sum_{\ell,m} \beta_i(\ell) \left( \int d\Omega_1 \Phi^*_{m_1} \Phi_{m_4} Y_{\ell,m} \right) \left( \int d\Omega_2 \Phi^*_{m_2} \Phi_{m_3} Y^*_{\ell,m} \right) \tag{C.12}$$

$$= \sum_{\ell,m} \beta_i(\ell) \frac{e^{i\pi(2s+m-2m_4-2m_3+m_1+m_2)}}{4\pi} (2\ell+1)(2s+1)^2 \begin{pmatrix} \ell & s & s \\ 0 & s & -s \end{pmatrix}^2$$

$$\times \begin{pmatrix} \ell & s & s \\ m & m_4 & -m_1 \end{pmatrix} \begin{pmatrix} \ell & s & s \\ -m & m_3 & -m_2 \end{pmatrix},$$

where $\begin{pmatrix} j_1 & j_2 & j_3 \\ m_1 & m_2 & m_3 \end{pmatrix}$ is the $3j$-symbol.[14] Because of the properties of the $3j$-symbol the expression vanishes unless

$$0 \leq \ell \leq 2s, \tag{C.13a}$$

$$m = m_4 - m_1, \qquad m = m_2 - m_3 \quad \Rightarrow \quad m_1 + m_2 = m_3 + m_4. \tag{C.13b}$$

---

[14]Note that here and in the following, to simplify expressions, we will always assume $2s \in \mathbb{N}$.

Including these conditions, the integral reduces to

$$\mathcal{I}^{(i)} = \sum_{\ell=0}^{2s} \frac{\beta_i(\ell)}{4\pi} (-1)^{2s-m_4-m_2}(2\ell+1)(2s+1)^2 \begin{pmatrix} \ell & s & s \\ 0 & s & -s \end{pmatrix}^2$$
$$\times \begin{pmatrix} \ell & s & s \\ m_1-m_4 & m_4 & -m_1 \end{pmatrix} \begin{pmatrix} \ell & s & s \\ m_2-m_3 & m_3 & -m_2 \end{pmatrix} \delta_{m_i},$$
(C.14)

with $\delta_{m_i} \equiv \delta_{m_1+m_2,m_3+m_4}$. This expression can be further simplified using the identity

$$\begin{pmatrix} s_1 & k & s_4 \\ -m_1 & m_1-m_4 & m_4 \end{pmatrix} \begin{pmatrix} s_2 & k & s_3 \\ -m_2 & m_2-m_3 & m_3 \end{pmatrix} \delta_{m_i} = \sum_{y=0}^{2s}(2y+1)(-1)^{m_3-m_1-2m_4}$$
$$\times \begin{Bmatrix} y & s_1 & s_2 \\ k & s_3 & s_4 \end{Bmatrix} \begin{pmatrix} s_1 & s_2 & y \\ m_1 & m_2 & -m_1-m_2 \end{pmatrix} \begin{pmatrix} s_3 & s_4 & y \\ m_3 & m_4 & -m_3-m_4 \end{pmatrix} \delta_{m_i},$$
(C.15)

such that

$$\mathcal{I}^{(i)} = \frac{1}{2} \sum_{y=0}^{2s} V_{2s-y}(2y+1) \begin{pmatrix} s & s & y \\ m_1 & m_2 & -m_1-m_2 \end{pmatrix} \begin{pmatrix} s & s & y \\ m_4 & m_3 & -m_3-m_4 \end{pmatrix} \delta_{m_i},$$
$$V_{2s-y} \equiv \sum_{\ell=0}^{2s}(-1)^{2s+y}\frac{\beta_i(\ell)}{2\pi}(2\ell+1)(2s+1)^2 \begin{Bmatrix} y & s & s \\ \ell & s & s \end{Bmatrix} \begin{pmatrix} \ell & s & s \\ 0 & s & -s \end{pmatrix}^2,$$
(C.16)

with $\{\cdots\}$ the Wigner $6j$-symbol. Writing the expression in terms of $V_{2s-y}$, known as the Haldane pseudopotential [24], has the advantage that, for a fixed $\beta_i$, only few values of $y$ give a non vanishing contribution. In the cases of interest

$$\beta_0(\ell) \quad \longleftrightarrow \quad V_{2s-y} = \begin{cases} \frac{\lambda_0}{R^2}\frac{(2s+1)^2}{2\pi(4s+1)}, & y=2s, \\ 0, & \text{else,} \end{cases}$$
(C.17a)

$$\beta_1(\ell) \quad \longleftrightarrow \quad V_{2s-y} = \begin{cases} \frac{\lambda_1}{R^4}\frac{s(2s+1)^2}{2\pi(4s+1)}, & y=2s, \\ \frac{\lambda_1}{R^4}\frac{s(2s+1)^2}{2\pi(4s-1)}, & y=2s-1, \\ 0, & \text{else.} \end{cases}$$
(C.17b)

From these expressions we can easily derive the relation between the $V_i$ pseudopotentials and the interaction parameters

$$\frac{\lambda_0}{R^2} = \frac{2\pi}{(2s+1)^2}\left((4s+1)V_0 + (4s-1)V_1\right),$$
$$\frac{\lambda_1}{R^4} = \frac{2\pi}{(2s+1)^2} \cdot \frac{(4s-1)V_1}{s}.$$
(C.18)

Passing to the rotation generators defined in (B.6)

$$J_z = (s+1)\sum_{m_1,m_2} \mathbf{c}_{m_1}^\dagger \mathbf{c}_{m_2} \int d\Omega \cos\theta\, \Phi_{m_1}^* \Phi_{m_2},$$
$$J_\pm = (s+1)\sum_{m_1,m_2} \mathbf{c}_{m_1}^\dagger \mathbf{c}_{m_2} \int d\Omega\, e^{\pm i\phi}\sin\theta\, \Phi_{m_1}^* \Phi_{m_2}.$$
(C.19)

Using the fact that

$$\cos\theta = 2\sqrt{\frac{\pi}{3}}Y_{1,0}, \qquad e^{\pm i\phi}\sin\theta = \mp 2\sqrt{2\frac{\pi}{3}}Y_{1,\pm 1}, \tag{C.20}$$

together with the properties in (C.4), it is straightforward to get

$$J_z = \sum_{m=-s}^{s} m\, \mathbf{c}_m^\dagger \mathbf{c}_m, \tag{C.21a}$$

$$J_+ = \sum_{m=-s}^{s-1} \sqrt{(s-m)(s+m+1)}\, \mathbf{c}_{m+1}^\dagger \mathbf{c}_m, \tag{C.21b}$$

$$J_- = \sum_{m=-s+1}^{s} \sqrt{(s+m)(s-m+1)}\, \mathbf{c}_{m-1}^\dagger \mathbf{c}_m. \tag{C.21c}$$

So defined, the generators satisfy the $SO(3)$ algebra

$$[J_z, J_\pm] = \pm J_\pm, \qquad [J_+, J_-] = 2J_z. \tag{C.22}$$

As we have explained before, we don't need to define the generators of translations and special conformal transformations separately. But rather, it is sufficient to construct their sum

$$\Lambda_z \equiv P_z + K_z = 2 \cdot 2\sqrt{\frac{\pi}{3}}\int_{S^2} Y_{1,0}T^0{}_0,$$

$$\Lambda_\pm \equiv (P_1 + K_1) \pm i(P_2 + K_2) = \pm 2 \cdot 2\sqrt{\frac{2\pi}{3}}\int_{S^2} Y_{1,\pm 1}T^0{}_0. \tag{C.23}$$

Using $T^0{}_0 \to \mathcal{H}$ in (C.6)

$$\Lambda_j = \sum_{m_i} \widetilde{\mathcal{I}}_j^{(i)} \left( \mathbf{c}_{m_1}^\dagger \mathbf{c}_{m_4} \mathbf{c}_{m_2}^\dagger \mathbf{c}_{m_3} - \mathbf{c}_{m_1}^\dagger \sigma_z \mathbf{c}_{m_4} \mathbf{c}_{m_2}^\dagger \sigma_z \mathbf{c}_{m_3} \right) - h\Lambda_j^{(h)} = \begin{cases} \Lambda_z, & j = 0, \\ \Lambda_\pm, & j = \pm 1, \end{cases}$$

$$\widetilde{\mathcal{I}}_j^{(i)} = 4\sqrt{\frac{\pi}{3}}\alpha_j \sum_{\ell,m} \beta^{(i)}(\ell) \left( \int d\Omega_1 \Phi_{m_1}^* \Phi_{m_4} Y_{1,j} Y_{\ell,m} \right) \left( \int d\Omega_2 \Phi_{m_2}^* \Phi_{m_3} Y_{\ell,m}^* \right),$$

$$\Lambda_j^{(h)} = 4\sqrt{\frac{\pi}{3}}\alpha_j \sum_{m_1,m_2} \int d\Omega Y_{1,j} \Phi_{m_1}^* \Phi_{m_2} \mathbf{c}_{m_1}^\dagger \sigma_x \mathbf{c}_{m_2} \tag{C.24}$$

$$= \sqrt{\frac{4s(2s+1)}{(s+1)}}\alpha_j \sum_m (-1)^{j-m+s} \begin{pmatrix} 1 & s & s \\ j & m & -j-m \end{pmatrix} \mathbf{c}_{m+j}^\dagger \sigma_x \mathbf{c}_m,$$

where we have used the same conventions in (C.11) and we have introduced

$$\alpha_j = \begin{cases} \mp\sqrt{2}, & j = \pm 1, \\ 1, & j = 0. \end{cases} \tag{C.25}$$

To solve for $\widetilde{\mathcal{I}}_j^{(i)}$ we can use the properties in (C.4), together with the identity for the product of spherical harmonics

$$Y_{\ell_1,m_1}Y_{\ell_2,m_2} = \sqrt{\frac{(2\ell_1+1)(2\ell_2+2)}{4\pi}}\sum_{k_{\min}}^{\ell_1+\ell_2} \sqrt{2k+1}(-1)^{m_1+m_2}Y_{k,m1+m2} \tag{C.26}$$

$$\times \begin{pmatrix} \ell_1 & \ell_2 & k \\ 0 & 0 & 0 \end{pmatrix} \begin{pmatrix} \ell_1 & \ell_2 & k \\ m_1 & m_2 & -m_1-m_2 \end{pmatrix}.$$

with $k_{\min} = \max(|\ell_1 - \ell_2|, |m_1 + m_2|)$. Then

$$\widetilde{\mathcal{I}}_j^{(i)} = \frac{\alpha_j}{2\pi} \delta_{m_1+m_2,m_3+m_4+j} (-1)^{j+m_{12}} \sum_{\ell=0}^{2s} \sum_{k=k_{\min}}^{\ell+1} \beta_i (2s+1)^2 (2k+1)(2\ell+1) \begin{pmatrix} 1 & \ell & k \\ 0 & 0 & 0 \end{pmatrix} \quad \text{(C.27)}$$

$$\times \begin{pmatrix} k & s & s \\ 0 & s & -s \end{pmatrix} \begin{pmatrix} \ell & s & s \\ 0 & s & -s \end{pmatrix} \begin{pmatrix} k & s & s \\ m_{14} & m_4 & -m_1 \end{pmatrix} \begin{pmatrix} \ell & s & s \\ m_{23} & m_3 & -m_2 \end{pmatrix} \begin{pmatrix} 1 & \ell & k \\ j & m_{32} & m_{23}-j \end{pmatrix},$$

where $m_{ij} \equiv m_i - m_j$ and $k_{\min} = \max(|m_{14}|, |\ell - 1|)$. Similarly as we have done for the pseudopotential, we can rewrite this expression in an easier form by using twice the identity in (C.15)

$$\widetilde{\mathcal{I}}_j^{(i)} = \alpha_j (-1)^{-m_1} \sum_{x,y} V_{y,x} (2y+1)(2x+1) \begin{pmatrix} y & s & x \\ m_1-j & m_2 & j-m_1-m_2 \end{pmatrix}$$

$$\times \begin{pmatrix} s & s & x \\ m_3 & m_4 & -m_3-m_4 \end{pmatrix} \begin{pmatrix} s & 1 & y \\ -m_1 & j & m_1-j \end{pmatrix} \delta_{m_1+m_2,m_3+m_4+j},$$

$$\text{(C.28)}$$

$$V_{y,x} \equiv \sum_{\ell=0}^{2s} \sum_{k=\ell-1}^{\ell+1} \frac{\beta_i}{2\pi} (-1)^{k-\ell+y} (2k+1)(2\ell+1)(2s+1)^2$$

$$\times \begin{Bmatrix} x & y & s \\ \ell & s & s \end{Bmatrix} \begin{Bmatrix} y & \ell & s \\ k & s & 1 \end{Bmatrix} \begin{pmatrix} 1 & \ell & k \\ 0 & 0 & 0 \end{pmatrix} \begin{pmatrix} k & s & s \\ 0 & s & -s \end{pmatrix} \begin{pmatrix} \ell & s & s \\ 0 & s & -s \end{pmatrix}.$$

For a fixed $\beta_i(\ell)$ only few $V_{y,x}$ are non vanishing

$$\beta_0(\ell) \quad \longleftrightarrow \quad V_{y,x=2s} = \frac{\lambda_0}{2\pi R^2} \begin{cases} \frac{(-1)^s}{(4s+1)} \sqrt{\frac{s(2s+1)^3}{s+1}}, & y=s, \\ -\frac{(-1)^s}{(4s+1)} \sqrt{\frac{s(2s+1)^3}{(s+1)(2s+3)}}, & y=s+1, \\ 0, & \text{else}, \end{cases} \quad \text{(C.29a)}$$

$$\beta_1(\ell) \quad \longleftrightarrow \quad V_{y,x} = \frac{\lambda_1}{2\pi R^4} \begin{cases} -\frac{(-1)^s}{(4s+1)} \sqrt{\frac{s^3(2s+1)^3}{s+1}}, & y=s,\, x=2s, \\ \frac{(-1)^s}{(4s+1)} \sqrt{\frac{s(2s+1)^5}{(s+1)(2s+3)}}, & y=s+1,\, x=2s, \\ -\frac{(-1)^s}{4s-1} \sqrt{\frac{s^3(2s+1)^3}{s+1}}, & y=s,\, x=2s-1, \\ \frac{(-1)^s}{4s-1} \sqrt{\frac{s(2s+1)^5(2s-1)}{(s+1)(3+2s)(4s+1)}}, & y=s+1,\, x=2s-1, \\ 0, & \text{else}. \end{cases} \quad \text{(C.29b)}$$

The $\Lambda$s satisfy the algebra

$$[J_z, \Lambda_z] = 0, \qquad [J_z, \Lambda_+] = \Lambda_+, \qquad [J_z, \Lambda_-] = -\Lambda_-, \qquad \text{(C.30a)}$$

$$[J_+, \Lambda_z] = -\Lambda_+, \qquad [J_+, \Lambda_+] = 0, \qquad [J_+, \Lambda_-] = 2\Lambda_z, \qquad \text{(C.30b)}$$

$$[J_-, \Lambda_z] = \Lambda_-, \qquad [J_-, \Lambda_+] = -2\Lambda_z, \qquad [J_-, \Lambda_-] = 0. \qquad \text{(C.30c)}$$

# D  Corrections to $(P + K)$ matrix elements from primaries

Our goal in this appendix is to obtain formulae from conformal perturbation theory for the matrix elements of $\mathcal{H}(x) - T_0^0$, specifically for the matrix elements of the form

$$\widetilde{F}_{\mathcal{O}_r T \mathcal{O}_n}^{(\text{state})}(x) \equiv \sum_{\mathcal{O}} g_{\mathcal{O}} \widetilde{F}_{\mathcal{O}_r T \mathcal{O}_n; \mathcal{O}}^{(\text{state})}(x), \qquad \text{(D.1)}$$

which are the contributions from a single operator to the quantities as defined in (24).

Consider the case where $\mathcal{O}_r$ is the identity operator and $\mathcal{O}_n$ is in the conformal multiplet of some primary operator we will call $\mathcal{O}$. In this case, the correlator that we need is the three-point function $\langle \mathcal{O}\mathcal{O}T \rangle$ on $S^{d-1} \times \mathbb{R}$, which is fixed by conformal invariance and the Ward identity to be [25]

$$\langle T^{00}(t_1, \widehat{x}_1)\mathcal{O}(t_2, \widehat{x}_2)\mathcal{O}(t_3, \widehat{x}_3)\rangle_{S^{d-1}\times\mathbb{R}}$$

$$= \frac{d\Delta}{(d-1)S_d} \frac{1}{d_{12}^d d_{23}^{2\Delta-d} d_{31}^d} \left( \frac{(\frac{d_{13}}{d_{12}}(\widehat{x}_1 \cdot \widehat{x}_2 - e^{t_1-t_2}) - \frac{d_{12}}{d_{13}}(\widehat{x}_1 \cdot \widehat{x}_3 - e^{t_1-t_3})^2}{d_{23}^2} - \frac{1}{d} \right), \quad \text{(D.2)}$$

where $S_d = \text{vol}(S^{d-1}) = \frac{2\pi^{d/2}}{\Gamma(d/2)}$, and

$$d_{ij}^2 \equiv 2(\cosh(t_i - t_j) - u_{ij}), \qquad u_{ij} \equiv \widehat{x}_i \cdot \widehat{x}_j. \quad \text{(D.3)}$$

We will just look at the cases where $\mathcal{O}_n$ is at most a level 2 descendant, which means we can expand in powers of $x_3$ up to $x_3^2$ – equivalently, we expand in powers of $e^{t_3}$ and keep up to $e^{(\Delta+2)t_3} = r_3^{\Delta+2}$ (where $r \equiv e^t$). To warm up, begin with just the leading power $e^{\Delta t_3} = r_3^{\Delta}$, which corresponds to taking $\mathcal{O}_n = \mathcal{O}$. Taking $d = 3$,

$$\langle T^{00}(t_1, \widehat{x}_1)\mathcal{O}(t_2, \widehat{x}_2)\mathcal{O}(t_3, \widehat{x}_3)\rangle_{S^2\times\mathbb{R}} \supset \frac{3\Delta}{2S_3} r_3^{\Delta} \left( \frac{r_2^{3-\Delta}\left(r_1^2\left(3u_{12}^2 - 1\right) - 4r_2 r_1 u_{12} + 2r_2^2\right)}{3\left(-2r_2 r_1 u_{12} + r_1^2 + r_2^2\right)^{5/2}} \right). \quad \text{(D.4)}$$

First, do the $\widehat{x}_2$ integral, to find

$$\langle T^{00}(0, \widehat{x}_1) \int d^2\Omega_2 \mathcal{O}(t_2, \widehat{x}_2)|\mathcal{O}\rangle_{S^2} = \frac{3\Delta}{2S_3}\left(\frac{1}{r_2}\right)^{\Delta} \begin{cases} \frac{8\pi}{3}, & r_2 > 1, \\ 0, & r_2 < 1. \end{cases} \quad \text{(D.5)}$$

Finally, integrate $\int_{-\infty}^{\infty} dt_E \cong \int_0^{\infty} \frac{dr_2}{r_2}$:

$$\widetilde{F}_{1T\mathcal{O};\mathcal{O}}^{(\text{state})}(x) = -g_{\mathcal{O}} \int_1^{\infty} \frac{dr_2}{r_2} \frac{3\Delta}{2S_3}\left(\frac{1}{r_2}\right)^{\Delta} \frac{8\pi}{3} = -g_{\mathcal{O}}. \quad \text{(D.6)}$$

Putting it all together,

$$\widetilde{F}_{1T\mathcal{O};\mathcal{O}}(x) = \langle\text{vac}|T^{00}(x)|\mathcal{O}_n\rangle + g_{\mathcal{O}}\langle\text{vac}|\mathcal{O}(x)|\mathcal{O}\rangle + \widetilde{F}_{1T\mathcal{O};\mathcal{O}}^{(\text{state})}(x)$$

$$= 0 + g_{\mathcal{O}} - g_{\mathcal{O}} = 0, \quad \text{(D.7)}$$

as it should be since $\langle\widetilde{\text{vac}}|\mathcal{H}(x)|\widetilde{\mathcal{O}}\rangle$ is rotationally invariant (both external states are scalars) and so it is equal to its average over the sphere, which is therefore just the matrix element of the Hamiltonian $\widetilde{H}$ between two different energy eigenstates.

Now, we can easily systematize this calculation. We can take $\vec{x} = \widehat{z}$ without loss of generality, since its dependence can be restored by considering the symmetries of the external state. We will also restrict to states $|\mathcal{O}_n\rangle$ with $j_z = 0$ without loss of generality. Then, at levels 0,1 and 2, all descendant states can be labeled by their $J$ value. Let $\mathcal{O}_{n,J}$ denote the level $n$ spin $J$

state. We find

$$\widetilde{F}^{(\text{state})}_{1T\mathcal{O}_{0,0};\mathcal{O}} = -1\,,$$

$$\widetilde{F}^{(\text{state})}_{1T\mathcal{O}_{1,1};\mathcal{O}} = -\frac{\sqrt{2}(3+\Delta+\Delta^2)}{(2+\Delta)\sqrt{3\Delta}}\,,$$

$$\widetilde{F}^{(\text{state})}_{1T\mathcal{O}_{2,0};\mathcal{O}} = -\frac{\sqrt{\Delta(2\Delta-1)}}{\sqrt{3}}\,,$$

$$\widetilde{F}^{(\text{state})}_{1T\mathcal{O}_{2,2};\mathcal{O}} = -\sqrt{\frac{1}{15}}\frac{2\left(\Delta^5+3\Delta^4+10\Delta^3+12\Delta^2+4\Delta-36\right)}{(\Delta-1)(\Delta+2)(\Delta+4)\sqrt{\Delta(\Delta+1)}}\,. \tag{D.8}$$

We also find by similar manipulations starting with the two-point function $\langle\mathcal{O}\mathcal{O}\rangle$ that the contributions from the shifts in the operators are, for a primary operator $\mathcal{O}$,

$$\widetilde{F}^{(\text{op})}_{1T\mathcal{O}_{0,0};\mathcal{O}}(\widehat{\Omega}) = 1\,,$$

$$\widetilde{F}^{(\text{op})}_{1T\mathcal{O}_{1,1};\mathcal{O}}(\widehat{\Omega}) = -\sqrt{\frac{2\Delta}{3}}\cos\theta\,,$$

$$\widetilde{F}^{(\text{op})}_{1T\mathcal{O}_{2,0};\mathcal{O}}(\widehat{\Omega}) = \frac{\sqrt{\Delta(2\Delta-1)}}{\sqrt{3}}\,,$$

$$\widetilde{F}^{(\text{op})}_{1T\mathcal{O}_{2,2};\mathcal{O}}(\widehat{\Omega}) = -2\sqrt{\frac{\Delta(\Delta+1)}{15}}\frac{3\cos^2\theta-1}{2}\,, \tag{D.9}$$

where we have reintroduced the $\Omega$ dependence based on the spin of the ket state (this just amounts to multiplying by $\frac{P_\ell(\cos\theta)}{P_\ell(1)}$ where $\ell$ is the spin of the ket; there is no $\phi$ dependence since by definition the ket state has $j_z = 0$ state). Therefore the combined contribution for a primary operator $\mathcal{O}$ is

$$\widetilde{F}_{1T\mathcal{O}_{0,0};\mathcal{O}}(\widehat{\Omega}) = 0\,,$$

$$\widetilde{F}_{1T\mathcal{O}_{1,1};\mathcal{O}}(\widehat{\Omega}) = \sqrt{\frac{2}{3\Delta}}\frac{\Delta-3}{\Delta+2}\cos\theta\,,$$

$$\widetilde{F}_{1T\mathcal{O}_{2,0};\mathcal{O}}(\widehat{\Omega}) = 0\,,$$

$$\widetilde{F}_{1T\mathcal{O}_{2,2};\mathcal{O}}(\widehat{\Omega}) = \frac{2\sqrt{\frac{3}{5}}(\Delta-3)(\Delta^3+2\Delta^2-4)}{\sqrt{\Delta(\Delta+1)}(\Delta-1)(\Delta+2)(\Delta+4)}\frac{3\cos^2\theta-1}{2}\,. \tag{D.10}$$

We can also easily read off the contribution to $\widetilde{F}_{1T,\mathcal{O}_{n,\ell};\mathcal{O}}$ from scalar descendant states by taking derivatives. We mainly are interested in the descendant operator $\nabla^2_{S^2}\mathcal{O}$, where

$$\nabla^2_{S^2}f(u) = -2uf'(u)+(1-u^2)f''(u)\,, \qquad u = \cos\theta\,. \tag{D.11}$$

Since $\nabla^2_{S^2} P_\ell(u) = -\ell(\ell+1)P_\ell(u)$, taking $\nabla^2_{S^2}$ just amounts to multiplication by $-\ell(\ell+1)$ where $\ell$ is the spin of the ket state. So

$$\widetilde{F}^{(op)}_{1T\mathcal{O}_{0,0};\nabla^2_{S^2}\mathcal{O}}(\widehat{\Omega}) = 0\,,$$

$$\widetilde{F}^{(op)}_{1T\mathcal{O}_{1,1};\nabla^2_{S^2}\mathcal{O}}(\widehat{\Omega}) = -2\sqrt{\frac{2\Delta}{3}}\cos\theta\,,$$

$$\widetilde{F}^{(op)}_{1T\mathcal{O}_{2,0};\nabla^2_{S^2}\mathcal{O}}(\widehat{\Omega}) = 0\,, \tag{D.12}$$

$$\widetilde{F}^{(op)}_{1T\mathcal{O}_{2,2};\nabla^2_{S^2}\mathcal{O}}(\widehat{\Omega}) = 12\sqrt{\frac{\Delta(\Delta+1)}{15}}\frac{3\cos^2\theta - 1}{2}\,.$$

The main quantity of interest is

$$\int d^2\Omega\cos\theta\,\widetilde{F}_{1T\mathcal{O}_{1,1}}(\widehat{x}) = \frac{4\pi}{3}\widetilde{F}_{1T,\mathcal{O}_{1,1}}(\widehat{z})$$

$$= g_\mathcal{O}\sqrt{\frac{2}{3\Delta}}\frac{\Delta-3}{\Delta+2}\frac{4\pi}{3} + \sum_{n=1}^{\infty}(-2)^n g_{\nabla^{2n}_{S^2}\mathcal{O}}\frac{4\pi}{3}\sqrt{\frac{2\Delta}{3}}\,, \tag{D.13}$$

where $\Delta = \Delta_\mathcal{O}$. No operators outside of the $\mathcal{O}$ conformal representation contribute, due to the fact that $\langle\text{vac}|\mathcal{O}'|\mathcal{O}\rangle = 0$ when $\mathcal{O}$ and $\mathcal{O}'$ are not in the same representation, and $\langle\text{vac}|T\{\int d^2\Omega\mathcal{O}'(x)T^{00}(\widehat{z})\}|\mathcal{O},\ell = 1\rangle \sim \langle\text{vac}|T\{\int d^2\Omega\mathcal{O}'(x)(K + P)\}|\mathcal{O},\ell = 1\rangle = 0$, since $T^{00}$ turns into $\Lambda$ by conservation of angular momentum, and then again the result vanishes if $\mathcal{O}'$ and $\mathcal{O}$ are not in the same conformal representation.

# E   Matrix elements of $P_A$ in CFT limit

Conceptually, computing the matrix elements of $P_A$ between states in the CFT limit is a straightforward application of the conformal algebra, though in practice the amount of effort involved in performing the necessary algebraic manipulations can be greatly reduced by organizing it effectively. It is typically easier to work with the algebra using an index-free notation, where it can be written as follows:

$$\left[D, J(y,y')\right] = 0\,, \qquad\qquad\qquad [K(z),P(y)] = 2z\cdot yD + 2J(z,y)\,,$$

$$[D, P(z)] = P(z)\,, \qquad\qquad\qquad [D, K(z)] = -K(z)\,,$$

$$[J(y,y'),P(z)] = z\cdot yP(y') - z\cdot y'P(y)\,, \qquad [J(y,y'),K(z)] = z\cdot yK(y') - z\cdot y'K(y)\,,$$

$$[J(y,y'),J(z,z')] = (z'\cdot yJ(z,y') + z\cdot y'J(z',y) - z'\cdot y'J(z,y) - z\cdot yJ(z',y'))\,. \tag{E.1}$$

The indices are all contracted with auxiliary vectors, so $K(z) \equiv z^A K_A$, $P(z) \equiv z^A P_A$, and $J(y,y') \equiv y^A y'^B J_{AB}$. We use a convention where $D$ is Hermitian, and acting on primaries $D|\text{prim}\rangle = \Delta|\text{prim}\rangle$.

For scalar primaries, we can obtain a closed-form expression for the matrix elements of $P_A$ by taking explicit simple integrals of CFT two-point functions. The basic idea is to start with the CFT two-point function

$$\langle(R\mathcal{O}R)(z)\mathcal{O}(y)\rangle = (1 - 2z\cdot y + z^2 y^2)^{-\Delta}\,, \tag{E.2}$$

where $R\mathcal{O}R$ is the conformal inversion of $\mathcal{O}$. Since conformal inversions exchange $P$ and $K$, we can act with $K$ on the bra state $\langle\mathcal{O}|$ by taking derivatives with respect to $x$ of the equation above. All descendant states of $\mathcal{O}$ can be obtained by repeatedly acting with $P_A$, so any matrix element of $P$ between any two descendant states can be written in terms of

$$\langle\mathcal{O}|K^{n'}(z)P(x)P^n(y)|\mathcal{O}\rangle = \frac{1}{n+1}x^A\frac{\partial}{\partial y^A}\langle\mathcal{O}|K^{n'}(z)P^{n+1}(y)|\mathcal{O}\rangle. \tag{E.3}$$

But because $P$ and $K$ implement translations on $\mathcal{O}$ and $R\mathcal{O}R$ respectively,

$$\langle(R\mathcal{O}R)(z)\mathcal{O}(y)\rangle = \sum_{n=0}^{\infty}\frac{1}{(n!)^2}\langle\mathcal{O}|K^n(z)P^n(y)|\mathcal{O}\rangle, \tag{E.4}$$

so by the standard generating function expression for Gegenbauer polynomials $C_n^{(\Delta)}$, we have

$$\langle\mathcal{O}|K^n(z)P^n(y)|\mathcal{O}\rangle = (n!)^2(y^2z^2)^{n/2}C_n^{(\Delta)}\left(\frac{y\cdot z}{\sqrt{y^2z^2}}\right). \tag{E.5}$$

To create states of a definite level $n$ and spin $\ell$, we can fix $n$ in this expression and integrate against spherical harmonics of $y, z$ to pick out the spin of the in and out state, respectively. In radial quantization, the spatial surfaces correspond to fixed radius in flat space coordinates, so we can set $y^2 = z^2 = 1$. However, we also want to include a factor of $P(x)$ as in (E.3), in which case we differentiate with respect to $y$ before setting $y^2 = 1$:

$$\langle\mathcal{O}|K^n(\widehat{z})P^n(\widehat{y})|\mathcal{O}\rangle = (n!)^2 C_n^{(\Delta)}(\widehat{y}\cdot\widehat{z}),$$

$$\langle\mathcal{O}|K^{n+1}(\widehat{z})P(x)P^n(\widehat{y})|\mathcal{O}\rangle = (n+1)!n!x^A\Big((2\Delta)(\widehat{z}^A - \widehat{y}^A\widehat{y}\cdot\widehat{z})C_n^{(\Delta+1)}(\widehat{y}\cdot\widehat{z}) \tag{E.6}$$

$$+ (n+1)\widehat{y}^A C_{n+1}^{(\Delta)}(\widehat{y}\cdot\widehat{z})\Big).$$

At level $n$, we can project onto the state with $j_z = 0$ and spin $\ell$ by integrating against the spherical harmonic $Y_{\ell 0}(\widehat{y}) \propto P_\ell(\cos\theta_y)$ (where the coordinate system is $\widehat{y} = (\sin\theta_y\cos\phi_y, \sin\theta_y\sin\phi_y, \cos\theta_y)$). Integrating against $\langle\mathcal{O}|K^n(\widehat{z})P^n(\widehat{y})|\mathcal{O}\rangle$ gives us the norm of the state:

$$|n,\ell\rangle = \frac{1}{N_{n,\ell}}\int d^2\Omega P_\ell(\cos\theta)\mathcal{O}(\widehat{y})|\text{vac}\rangle,$$

$$N_{n,\ell}^2 = \int d^2\Omega_y d^2\Omega_z P_\ell(\cos\theta_y)P_\ell(\cos\theta_z)\langle\mathcal{O}|K^n(\widehat{z})P^n(\widehat{y})|\mathcal{O}\rangle \tag{E.7}$$

$$= \frac{2^{2\Delta-1}\Gamma(n+1)^2\Gamma\left(\frac{1}{2}(n-\ell-1)+\Delta\right)\Gamma\left(\frac{\ell+n}{2}+\Delta\right)}{(2\ell+1)\Gamma(2\Delta-1)\Gamma\left(\frac{1}{2}(n-\ell+2)\right)\Gamma\left(\frac{1}{2}(\ell+n+3)\right)},$$

and then integrating against $\langle\mathcal{O}|K^{n+1}(\widehat{z})P(\widehat{e}_z)P^n(\widehat{y})|\mathcal{O}\rangle$ with $\widehat{e}_z = (0,0,1)$ gives us the matrix element:

$$\langle n+1,\ell'|P_z|n,\ell\rangle = \frac{\int d^2\Omega_y d^2\Omega_z P_{\ell'}(\cos\theta_z)P_\ell(\cos\theta_y)\langle\mathcal{O}|K^{n+1}(\widehat{z})P(\widehat{e}_z)P^n(\widehat{y})|\mathcal{O}\rangle}{N_{n+1,\ell'}N_{n,\ell}}$$

$$= \begin{cases} \sqrt{\frac{\ell^2(n-\ell+2)(2\Delta+n-\ell-1)}{(2\ell+1)(2\ell-1)}} & (\ell' = \ell-1), \\ \sqrt{\frac{(\ell+1)^2(n+\ell+3)(2\Delta+n+\ell)}{(2\ell+1)(2\ell+3)}} & (\ell' = \ell+1), \\ 0, & \text{else.} \end{cases} \tag{E.8}$$

As a check, one can verify that

$$\left(\sum_{\ell'=\ell\pm1}|\langle n+1,\ell'|P_z|n,\ell\rangle|^2\right)-\left(\sum_{\ell'=\ell\pm1}|\langle n,\ell|P_z|n-1,\ell'\rangle|^2\right)=2(\Delta+n),\qquad\text{(E.9)}$$

as required by the commutator $[K_z,P_z]=2D$. For a more complicated check, one can evaluate the Casimir, $\mathcal{C}=D^2+J^2-\frac{1}{2}\{K_A,P_A\}$. We can relate matrix elements of $K_AP_A$ and $P_AK_A$ to those of $K_zP_z$ and $P_zK_z$ respectively using the Wigner-Eckart theorem, so in terms of the matrix elements for $P_z$ the Casimir acting on a state of level $n$ and spin $\ell$ evaluates to

$$\langle n,\ell|\mathcal{C}|n,\ell\rangle=(\Delta+n)^2+\ell(\ell+1)$$
$$-\frac{1}{2}\Big(\frac{2\ell+3}{\ell+1}|\langle n+1,\ell+1|P_z|n,\ell\rangle|^2+\frac{2\ell-1}{\ell}|\langle n+1,\ell-1|P_z|n,\ell\rangle|^2$$
$$+\frac{2\ell+3}{\ell+1}|\langle n,\ell|P_z|n-1,\ell+1\rangle|^2+\frac{2\ell-1}{\ell}|\langle n,\ell|P_z|n-1,\ell-1\rangle|^2\Big)$$
$$=\Delta(\Delta-3),$$

(E.10)

as expected. In fact, because for each state there are only two nonzero matrix elements of $P_z$ connecting the state to states at lower levels and two connecting it to states at higher levels, one can reverse the logic and with only these two consistency relations one has a recursion relation between matrix elements at neighboring levels that one can solve to find all the values of $\langle n+1,\ell'|P_z|n,\ell\rangle$, providing an independent proof of (E.8).

For the case of spinning primaries, it is easier to use such recursion relations to determine the matrix elements of $P$. Consider a primary with dimension $\Delta$ and spin $\ell_0$, and introduce the shorthand

$$Z_{n,\ell}^{\pm}\equiv|\langle n+1,\ell\pm1|P_z|n,\ell\rangle|^2,\qquad\text{(E.11)}$$

for the sum over matrix-element-squareds of $P_z$ between level $n$ spin $\ell$ descendants and level $n+1$ spin $\ell\pm1$ descendants, still in the $j_z=0$ sector. The commutator $[K_z,P_z]=2D$ evaluated in the $j_z=0$ component at level $n$ and spin $\ell$ implies

$$Z_{n,\ell}^{+}+Z_{n,\ell}^{-}-Z_{n-1,\ell-1}^{+}-Z_{n-1,\ell+1}^{-}=2(\Delta+n)N(n,\ell),\qquad\text{(E.12)}$$

where $N(n,\ell)$ is the number of descendants at level $n$ and spin $\ell$, which for generic $\Delta$ (i.e. for $\Delta\neq\ell_0+1$) can be written $N(n,\ell)=\sum_{j=0}^{\lfloor\frac{n}{2}\rfloor}\Theta(\ell+\ell_0\geq n-2j\geq|\ell-\ell_0|)$. The matrix elements of $P_z$ vanish between descendants with the same spin $\ell$ in the $j_z=0$ sector, but they arise in other $j_z$ sectors. If we evaluate the same commutator in the $j_z=1$ sector,

$$\frac{\ell(\ell+2)}{(\ell+1)^2}(Z_{n,\ell}^{+}-Z_{n-1,\ell+1}^{-})+\frac{(\ell+1)(\ell-1)}{\ell^2}(Z_{n,\ell}^{-}-Z_{n-1,\ell-1}^{+})+Z_{n,\ell}^{01}-Z_{n-1,\ell}^{01}=2(\Delta+n)N(n,\ell),$$

(E.13)

where

$$Z_{n,\ell}^{01}\equiv|\langle n+1,\ell;j_z=1|P_z|n,\ell;j_z=1\rangle|^2,\qquad\text{(E.14)}$$

again implicitly summed over all descendants with the indicated quantum numbers. To obtain another recursion formula, we can look at the commutator $[K_-,P_+]=2(D+J_z)$, now in the highest-weight ($j_z=\ell$) component at level $n$ and spin $\ell$:

$$Z_{n,\ell}^{+}\frac{2\ell+1}{\ell+1}-\ell Z_{n-1,\ell}^{01}-Z_{n-1,\ell-1}^{+}\frac{2\ell-1}{\ell}-Z_{n-1,\ell+1}^{-}\frac{1}{(\ell+1)^2}=2(\Delta+n+\ell)N(n,\ell),\qquad\text{(E.15)}$$

where we have used the Wigner-Eckart theorem to relate matrix elements of $P_+$ in the highest-weight components to the matrix elements of $P_z$ in the $j_z = 0$ component. Solving these three equations, we obtain the following recursion relation:

$$
Z_{n,\ell}^+ = \frac{2(\ell+1)(\Delta+\ell+n)}{2\ell+1}N(n,\ell) + \frac{1}{(\ell+1)(2\ell+1)}Z_{n-1,\ell+1}^- + \frac{(\ell+1)(2\ell-1)}{\ell(2\ell+1)}Z_{n-1,\ell-1}^+
$$
$$
+ \frac{\ell(\ell+1)}{2\ell+1}Z_{n-1,\ell}^{01},
$$
$$
Z_{n,\ell}^- = \frac{2\ell(\Delta+n-\ell-1)}{2\ell+1}N(n,\ell) + \frac{\ell(2\ell+3)}{(\ell+1)(2\ell+1)}Z_{n-1,\ell+1}^- + \frac{1}{\ell(2\ell+1)}Z_{n-1,\ell-1}^+ \qquad \text{(E.16)}
$$
$$
- \frac{\ell(\ell+1)}{2\ell+1}Z_{n-1,\ell}^{01},
$$
$$
Z_{n\ell}^{01} = 2(\Delta+n)N(n,\ell) - \frac{(\ell+1)(\ell-1)}{\ell^2}(Z_{n,\ell}^- - Z_{n-1,\ell-1}^+) + \frac{\ell(\ell+2)}{(\ell+1)^2}(Z_{n-1,\ell+1}^- - Z_{n,\ell}^+)
$$
$$
+ Z_{n-1,\ell}^{01}.
$$

Combined with the boundary conditions that

$$
Z_{n,\ell} = 0, \text{ if } n < 0, \text{ or } \ell < 0, \text{ or } |\ell-\ell_0| > n, \tag{E.17}
$$

this completely determines the values of the $Z_{n,\ell}$s.

## F  Useful OPE data

Table 1: $\mathbb{Z}_2$-even operators with conformal dimension $\Delta < 7$. Unless stated otherwise, the data are taken from [26], which extended previous results in [27].

| $\mathcal{O}$ | $\Delta$ | $\ell$ | $f_{\epsilon\epsilon\mathcal{O}}$ | $f_{\epsilon'\epsilon'\mathcal{O}}$ |
|---|---|---|---|---|
| $\epsilon$ | 1.412625(10) | 0 | 1.532435(19) | 2.3956 [19, 28] |
| $T_{\mu\nu}$ | 3 | 2 | 0.8891471(40) | |
| $\epsilon'$ | 3.82951(61) [29] | 0 | 1.5362(12) [29] | 7.6771 |
| $C_{\mu\nu\rho\sigma}$ | 5.022665(28) | 4 | 0.24792(20) | |
| $T'_{\mu\nu}$ | 5.50915(44) | 2 | 0.69023(49) | |
| $C'_{\mu\nu\rho\sigma}$ | 6.42065(64) | 4 | −0.110247(54) | |
| $\epsilon''$ | 6.8956(43) | 0 | 0.1279(17) | |

Table 2: $\mathbb{Z}_2$-odd operators with conformal dimension $\Delta < 7$. Unless stated otherwise, the data are taken from [26], which extended previous results in [27].

| $\mathcal{O}$ | $\Delta$ | $\ell$ |
|---|---|---|
| $\sigma$ | 0.5181489(10) | 0 |
| $\sigma_{\mu\nu}$ | 4.180305(18) | 2 |
| $\sigma_{\mu\nu\rho}$ | 4.63804(88) | 3 |
| $\sigma'$ | 5.262(89) [29] | 0 |
| $\sigma_{\mu\nu\rho\sigma}$ | 6.112674(19) | 4 |
| $\sigma_{\mu\nu\rho\sigma\delta}$ | 6.709778(27) | 5 |
| $\sigma'_{\mu\nu}$ | 6.9873(53) | 2 |

## G   Estimates for primary conformal dimensions

In this appendix, we report the estimate of conformal dimensions computed at $N = 16$ for the primary operators identified using the procedure in Sec. 5.3 in the $\mathbb{Z}_2$-even and $\mathbb{Z}_2$-odd sectors. The errors can be inferred from comparison with the bootstrap results in Appendix F, which are know to very high accuracy.

Table 3: Conformal dimensions for $\mathbb{Z}_2$-even primary operators at $N = 16$.

| $\mathcal{O}$ | $\Delta$ | $\ell$ |
|---|---|---|
| $\epsilon$ | 1.402623 | 0 |
| $T_{\mu\nu}$ | 3.005017 | 2 |
| $\epsilon'$ | 3.80967 | 0 |
| $C_{\mu\nu\rho\sigma}$ | 5.121876 | 4 |
| $T'_{\mu\nu}$ | 5.550384 | 2 |

Table 4: Conformal dimensions for $\mathbb{Z}_2$-odd primary operators at $N = 16$.

| $\mathcal{O}$ | $\Delta$ | $\ell$ |
|---|---|---|
| $\sigma$ | 0.5188153 | 0 |
| $\sigma_{\mu\nu}$ | 4.212098 | 2 |
| $\sigma_{\mu\nu\rho}$ | 4.613442 | 3 |
| $\sigma'$ | 5.265805 | 0 |
| $\sigma_{\mu\nu\rho\sigma}$ | 6.209049 | 4 |

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
