# Peer review of "Constructing the Infrared Conformal Generators on the Fuzzy Sphere"

_SciPost Physics, doi:SciPost Phys. 18, 086 (2025)_

## Round 1 · Referee Report · Anonymous (Referee 1) · 2024-11-28

Report

The authors use the recently developed fuzzy sphere regularization method to study the emergence of conformal symmetry at criticality. In particular, they realize numerically, in terms of the microscopic fuzzy sphere description, the CFT generators. This generalizes to 3D CFTs earlier work by other authors carried out in quantum critical spin chains, which describe 2D CFTs. The benefit of carrying out such an exercise, even though one already has access to scaling dimensions and OPE coefficients from earlier fuzzy sphere work, is that the authors can systematically test which operators are descendants and which are primaries. While potentially redundant for low-lying states in the spectrum, this is a crucial for states higher up in the spectrum. Phrased in a more pedestrian way, we would like to have a criterion separate from integer spacing of scaling dimensions for singling out primaries, especially given the existence of finite size effects that can move operators around in the spectrum. A key point of the analysis is that the critical point is reached by tuning specific finite size corrections (due to specific operators) to be as small as possible. This requires input from the known Ising CFT, i.e. the present analysis is not independent of earlier results; nevertheless, this has the potential upside of providing more sharp results.

The paper itself is very well written, and computations are made explicit to the extent possible (which I appreciate) and are supplemented by extensive appendices. The authors show thorough control of the subject matter, and the writing is clear and pedagogical without becoming verbose. Several useful auxiliary results are also provided.

As far as the goal of identifying primaries is concerned, the work is partially successful: the authors do indeed succeed in identifying some primaries in a robust way. That being said, all operators that were unambiguously identified as primaries are low-lying operators (up to dimension roughly 6). This is somewhat disappointing, given a) the amount of input about the theory used to obtain them and b) the fact that the Ising CFT presents the best case scenario, i.e. it is the cheapest non-trivial CFT to study numerically. This raises doubts as to what extent these methods could be applied to other theories and obtain strong results, given that already in the Ising CFT and with a respectable maximum system size, the results are mild. I should emphasize: this is not the fault of the authors, who have carried out a thorough analysis from start to end.

Below I mention a few comments that I would like to be addressed:

1) Reference 24 should mention something along the lines of "unpublished work", "work in progress", "private communication", etc (currently there is nothing).

2) There might be a typo in 3.11: I think the power of $N$ should be $(3-\Delta_O)/2$ instead of $(\Delta_O-3)/2$. As it stands now, the contribution of irrelevant operators at $N \rightarrow \infty $ doesn't go to zero.

3) In figure 10, the caption mentions "A vertical line indicates roughly where the cutoff appears to be by eye". However, already left of the vertical line there are states that have a negative dimension in the y-axis. At my most optimistic, I wouldn't put the vertical line further right than Delta ∼ 4, at which point there are already deviations from the dashed diagonal line. Is this a typo? The same goes for figure 11: the deviations from the dashed line are already large left of the vertical line.

Also, in the text on page 25 it mentions "However, as the dimension grows, the errors become large, and one finds negative values of [Kz,Pz] at ∆ ∼ 5.25, as indicated by a vertical line in the figure.". As mentioned above, I see negative values in fig 10 before Δ=5.25.

4) Given that the authors have performed a rather interesting tuning to reach the fixed point, whose aim is to minimize finite size contributions, I would like to encourage them to summarize the CFT data they have obtained in this work in a simple table (essentially the dimensions of the primaries). I think this will be a useful reference point for future readers.

5) This last point is more taste dependent, but I feel the paper ends a little abruptly. This could be solved, for example, by moving the last paragraph on page 3 to just above the acknowledgements and turning it into a discussion/outlook section, while also fleshing it out a little bit. The authors could also comment on the outlook of the technology developed for theories that are not the Ising model, where less is known.

To conclude, I consider the paper to be well written and to contain valid contributions to the field. Once the minor comments above have been addressed, I will be happy to recommend it for publication.

Requested changes

See points 1) through 5) in the report.

Recommendation

Ask for minor revision

---

## Round 1 · Referee Report · Anonymous (Referee 3) · 2024-12-11

Strengths

1- Clear exposition of the methodology. 2- Self-contained.

Report

This paper builds upon the fuzzy sphere regularization method introduced by Zhu et al. (ref. [6] in the paper) to compute conformal data of the 3D Ising CFT. The technique involves finding an appropriate effective Hamiltonian of interacting non-relativistic fermions on a cylinder with a background magnetic field and truncating the model to the lowest landau level (LLL). The spectrum of the corresponding truncated Hamiltonian serves as an approximation of the operator spectrum of the Ising CFT.  

The authors significantly advance the state of the art of these calculations by computing matrix elements of conformal generators in the truncated LLL model. This enables them to: 1) quantify how many states reliably capture CFT physics. 2) provide a systematic method to identify primary operators in the truncated spectrum.  In practice, these matrix elements are computed by tuning contributions from irrelevant operators in the conformal charges using input from conformal perturbation theory and light operator data from bootstrap calculations. 

The paper is technical, but the authors have done an excellent job of making the ideas and methodology accessible to an audience unfamiliar with fuzzy regularizations of CFTs. I particularly appreciated the concise review supplemented by the exhaustive appendices, which I find well-suited for a hep-th audience. The results are interesting and relevant, and provide a starting point for future investigations.

Requested changes

I have only one minor comment I would like the authors to address. As mentioned in the Introduction and Summary section, the general motivation behind the work is to access high-energy data of 3d Ising (or in general any CFT for which a fuzzy regularization can be established). This fully justifies the use of low-energy data from bootstrap calculations as input for the tuning procedure. However, I feel that the Summary section is missing some commentary on the final result in light of this initial motivation. 
More precisely: at the highest value of the cutoff $N=16$ one can reliably extract scaling dimensions only up to around $\Delta \sim 6$ (which are comparable with the heaviest operators measured by the bootstrap in ref [26] and reported in Appendix F). Is it then correct to conclude that these operators are not heavy enough to access the hydrodynamic and large-charge phases (the latter arising in some $O(N)$ model generalization of the setup) described in the references [1-4]? Is there a rough idea of the threshold “heaviness” required to observe the onset of these phases? 

Including even partial answers (if known) to these questions in the Summary would help tie back to the original motivation.

Recommendation

Ask for minor revision

---

## Round 1 · Referee Report · Connor Behan (Referee 2) · 2024-12-11

Report

The present paper makes important improvements to the fuzzy sphere regulator which is a numerical technique designed to complement the conformal bootstrap. A selling point is that it gives access to high energy states of some experimentally realizable CFTs in a way which makes essential use of the state operator map. To assess the health of the truncated spectrum, previous studies have largely focused on grouping the energies into families with integer spacing, but conformal invariance is a much stronger statement than this. By explicitly constructing conformal generators on the fuzzy sphere, the authors are able to carry out more detailed structural tests.

The authors also discuss the usefulness of these tests from a Wilsonian viewpoint. In particular, they convincingly demonstrate the fact that tuning relevant operators in order to reach a fixed point is not enough. The irrelevant operators should be tuned as well since it is numerically prohibitive to reach the deep IR where they are guaranteed to drop out. This leads to an interesting procedure for improving fuzzy sphere results. After tuning the Haldane pseudopoentials corresponding to primary operators in the Hamiltonian $H$, one can effectively tune them all over again in the operator $\Lambda_A = K_A + P_A$. What this actually corresponds to is a tuning of descendant operators in the Hamiltonian density which drop out of the Hamiltonian itself. With an optimized $\Lambda_A$ operator, its bimodal matrix elements can be used to estimate $K_A$ and $P_A$ individually and therefore recover conformal primaries which sometimes differ from energy eigenstates.

The main results are clearly interesting and there is also an effort to be pedagogical about some aspects that are sometiems underappreciated. As such, I have very few suggestions for the body of the paper but I think the appendices need a few more.

Requested changes

  1. Figure 2 has the typo "less then" and page 10 says "to try tune".
  2. Numbering both lines in equation (5.5), (5.6) might be unintentional. This is also done several times in Appendix C.
  3. When first mentioning $\tilde{\Lambda}$ it would be best to refer to it as the analogue of $\Lambda$ but based on the UV Hamiltonian density. Its definition in (4.12) only comes on the next page.
  4. Appendix B says "simply compute sum" and Appendix C says "only few values of y gives".
  5. It would be more clear to give the fermions explicit angle arguments in (C.5).
  6. Relatedly, (C.24) might lead some readers to think that the angles $\Omega_1$ and $\Omega_2$ are not on equal footing. Perhaps a sentence could be added to clarify this.
  7. One side of (E.4) has (x, y) arguments while the other has (z, y).

Finally, some things in Appendix D were a bit hard to follow. For one thing, (D.1) has some typos and it seems to already be specialized to the $O_n = O$ case even though the text mentions this simplification later. But I am mostly concerned with the subscripts on $\tilde{F}_{O_r, T, O_n}$. Once this correction to a 3pt matrix element is split into "state" and "op" pieces, both of them receive contributions from infinitely many fourth operators. So it would make sense to add a fourth subscript for "state" and "op" but Appendix D only does this for "op". Yet another subscript seems to be dropped underneath (D.9). I realize that the fourth subscript on "state" is technically redundant when $O_r = 1$ because this forces $O$ to be the primary in the conformal multiplet of $O_n$ but this explanation only comes at the end of Appendix D.

Recommendation

Ask for minor revision

---

## Round 2 · Referee Report · Connor Behan (Referee 2) · 2025-2-16

Report

The authors have recently made corrections and added statements which give more context to the results. I appreciate these and think that the paper is ready now.

Recommendation

Publish (easily meets expectations and criteria for this Journal; among top 50%)

---

## Round 2 · Referee Report · Anonymous (Referee 1) · 2025-2-18

Report

The authors have addressed a number of my comments, for which I thank them.

On page 25, in the paragraph under equation (5.2), they have rephrased the cutoff (which allowed for negative scaling dimensions, as can be seen in figure 10) as "optimistic". I want to state, for the record, that any cutoff that allows for negative scaling dimensions is more than just "optimistic".

That being said, given that the authors have addressed most of my complaints, I recommend the paper for publication.

Recommendation

Publish (meets expectations and criteria for this Journal)

---

## Round 2 · Referee Report · Anonymous (Referee 3) · 2025-2-20

Report

The authors addressed my question and mentioned explicitly an example of a model in which their study could be applied to match results obtained via EFT methods.

I believe this adds to their motivation for future research, and I recommend the paper for publication.

Recommendation

Publish (easily meets expectations and criteria for this Journal; among top 50%)

---

## Round 2 · List of Changes

We have fixed typos and extraneous equation numbers.
We have reviewed the notation in appendix D to make it consistent.
We have added clarification comments to address the referees' requests:
- We have added a comment in the summary about when high-dimension limits are expected to start to emerge
- We added a definition of $\tilde{\Lambda}$ on page 16, immediately after it first appears
-We have added an appendix summarizing the dimensions emerging from our analysis as suggested by the referees

---

## Editorial Decision

published